EMBO
Molecular Medicine

# An oligoclonal antibody durably overcomes resistance of lung cancer to third-generation EGFR inhibitors

Maicol Mancini[1], Hilah Gal[2], Nadège Gaborit[1], Luigi Mazzeo[1], Donatella Romaniello[1], Tomer Meir Salame[3], Moshit Lindzen[1], Georg Mahlknecht[1], Yehoshua Enuka[1] (ID), Dominick GA Burton[2], Lee Roth[1], Ashish Noronha[1], Ilaria Marrocco[1], Dan Adreka[1], Raya Eilam Altstadter[4], Emilie Bousquet[5], Julian Downward[6,7], Antonio Maraver[5], Valery Krizhanovsky[2] (ID) & Yosef Yarden[1,*] (ID)

## Abstract

Epidermal growth factor receptor (*EGFR*) mutations identify patients with lung cancer who derive benefit from kinase inhibitors. However, most patients eventually develop resistance, primarily due to the T790M second-site mutation. Irreversible inhibitors (e.g., osimertinib/AZD9291) inhibit T790M-EGFR, but several mechanisms, including a third-site mutation, C797S, confer renewed resistance. We previously reported that a triple mixture of monoclonal antibodies, 3×mAbs, simultaneously targeting EGFR, HER2, and HER3, inhibits T790M-expressing tumors. We now report that 3×mAbs, including a triplet containing cetuximab and trastuzumab, inhibits C797S-expressing tumors. Unlike osimertinib, which induces apoptosis, 3×mAbs promotes degradation of the three receptors and induces cellular senescence. Consistent with distinct mechanisms, treatments combining 3×mAbs plus sub-inhibitory doses of osimertinib synergistically and persistently eliminated tumors. Thus, oligoclonal antibodies, either alone or in combination with kinase inhibitors, might preempt repeated cycles of treatment and rapid emergence of resistance.

**Keywords** antibody therapy; apoptosis; kinase inhibitor; NSCLC; T790M
**Subject Categories** Cancer; Pharmacology & Drug Discovery; Respiratory System

## Introduction

Somatic mutations in the gene encoding the epidermal growth factor receptor (EGFR) are detected in approximately 12% of non-small cell lung cancers (NSCLCs) from Caucasian patients and in 30–40% of NSCLCs from Asian patients (Lynch *et al*, 2004; Paez *et al*, 2004; Pao *et al*, 2004). All oncogenic mutations locate to the catalytic, tyrosine-specific kinase domain of EGFR. Two aberrations, the L858R mutation and the exon 19 deletion (del746-750), represent the vast majority of activating EGFR mutations. Treatment of patients with gefitinib or erlotinib, two reversible tyrosine kinase inhibitors (TKIs), is associated with response rates that are superior to treatment with chemotherapy (Mok *et al*, 2009). However, despite initial drug activity, all patients acquire resistance within approximately 1 year (Mok *et al*, 2009; Rosell *et al*, 2009). The most common mechanism (> 50%) of acquired resistance involves a secondary mutation, T790M (Kobayashi *et al*, 2005; Pao *et al*, 2005; Oxnard *et al*, 2011).

Two strategies have been developed to overcome EGFR TKI resistance. The first involves synthesis of novel compounds (Zhou *et al*, 2009), including irreversible TKIs, which covalently conjugate to cysteine 797 of EGFR. For example, afatinib (Gilotrif™) covalently binds with EGFR and HER2 (Regales *et al*, 2009). However, inhibition of wild-type EGFR and relatively high drug doses required for inhibition of T790M-EGFR (Yap *et al*, 2010) limit afatinib application. Secondly, a combination of afatinib and cetuximab (an anti-EGFR monoclonal antibody) overcomes T790M-mediated resistance (Regales *et al*, 2009; Keating, 2014). A study that combined afatinib and cetuximab, and recruited patients who acquired TKI resistance, showed an overall response rate of 29%, but this result was comparable in T790M-positive and in T790M-negative tumors (Janjigian

1  Department of Biological Regulation, Weizmann Institute of Science, Rehovot, Israel
2  Department of Molecular Cell Biology, Weizmann Institute of Science, Rehovot, Israel
3  Department of Biological Services, Weizmann Institute of Science, Rehovot, Israel
4  Department of Veterinary Resources, Weizmann Institute of Science, Rehovot, Israel
5  Oncogenic Pathways in Lung Cancer, Institut de Recherche en Cancérologie de Montpellier (IRCM), Inserm U1194, Montpellier Cedex 5, France
6  Signal Transduction Laboratory, Francis Crick Institute, London, UK
7  Lung Cancer Group, The Institute of Cancer Research, London, UK
*Corresponding author. Tel: +972 8 934 3974; Fax: +972 8 934 2488; E-mail: yosef.yarden@weizmann.ac.il

  

*et al*, 2014; Pirazzoli *et al*, 2014). The pharmacological limitations of second-generation inhibitors might be overcome by the newest, third-generation inhibitors, such as osimertinib, CO-1686 (rocile-tinib), and HM61713 (reviewed in Costa & Kobayashi, 2015; Mancini & Yarden, 2015). Osimertinib inhibits T790M-EGFR, while sparing wild-type EGFR (Liao *et al*, 2015). In a phase I trial, osimertinib demonstrated manageable tolerability and 51% response rate among T790M mutant tumors (Janne *et al*, 2015). Subsequent studies not only demonstrated that osimertinib has significantly greater efficacy than chemotherapy (Mok *et al*, 2017), but also confirmed its ability to provide a high overall response rate, as well as confer durable response (Yang *et al*, 2017). Nevertheless, patients treated with osimertinib acquire resistance, due to several mechanisms, including emergence of C797S mutations (Eberlein *et al*, 2015; Thress *et al*, 2015). Other mechanisms include aberrant expression of NRAS and KRAS (Eberlein *et al*, 2015), or activation of MET and HER2, a kin of EGFR (Planchard *et al*, 2015).

Unlike the mutation-prone intracellular domain of EGFR, the extracellular domain is rarely mutated, which predicts low incidence of resistance to anti-EGFR monoclonal antibodies (mAbs). We previously examined this strategy and uncovered compensatory loops that increase transcription of HER2 and HER3 in response to an anti-EGFR mAb (Mancini *et al*, 2015). Preventing this by means of a triple combination of mAbs to EGFR, HER2 and HER3 (hereinafter, 3×mAbs) robustly inhibited tumor growth. The present study compares 3×mAbs and osimertinib. While in animal models, both treatments effectively inhibited growth of T790M-positive tumors, when tested *in vitro* and in animals osimertinib-induced apoptosis, whereas 3×mAbs caused cellular senescence and weak apoptosis. Importantly, prolonged exposure to osimertinib promoted emergence of resistant tumor cells, including C797S-EGFR expressing cells, which remained sensitive to 3×mAbs. Congruent with distinct mechanisms of action of 3×mAbs and osimertinib, treatments of tumor-bearing mice with 3×mAbs plus a sub-inhibitory dose of osimertinib durably prevented tumor relapses after ending all treatments. Taken together, these observations offer a new NSCLC treatment strategy, potentially able to overcome many, if not all resistance-conferring EGFR kinase mutations.

# Results

## Combining trastuzumab and cetuximab with an anti-HER3 antibody strongly inhibits erlotinib-resistant tumors

EGFR's intracellular part presents mutations responsible for recurring TKI resistance (Camidge *et al*, 2014), but the ectodomain is rarely mutated, such that anti-EGFR antibodies might overcome mutation-driven resistance to TKIs. We previously examined this scenario using two NSCLC cell lines (Mancini *et al*, 2015): patient-derived H1975 cells expressing a double mutant of EGFR, L858R, and T790M, and the PC9ER cell line, a derivative of PC9 (del746-750 EGFR), which acquired the T790M mutation (de Bruin *et al*, 2014). Inhibition of both animal models required simultaneous treatment with three homemade mAbs, to EGFR, HER2, and HER3 (Mancini *et al*, 2015). Similarly, we now report that combining two clinically approved mAbs, cetuximab (anti-EGFR) and trastuzumab (anti-HER2), with

a murine anti-HER3 (mAb33), partly inhibited *in vitro* growth of PC9ER and H1975 cells (Fig EV1A) and almost completely prevented tumorigenic growth of PC9ER cells in animals (Fig 1A). Moreover, this effect persisted at least 30 days post-treatment. In similarity to the murine anti-EGFR antibodies we previously tested (Mancini *et al*, 2015), cetuximab was more effective *in vivo* than singly applied anti-HER2 or anti-HER3 antibodies. In conclusion, the therapeutic activities of cetuximab and trastuzumab can be augmented by adding an anti-HER3 antibody, such that the oligoclonal mixture of two humanized antibodies and a murine mAb persistently inhibits TKI-resistant NSCLC models.

## Third-generation TKIs and 3×mAbs comparably inhibit TKI-resistant tumors, but their biological effects are distinct

Toward *in vivo* comparisons of 3×mAbs and a third-generation TKI, we examined effects on metabolic activity and EGFR phosphorylation. As predicted, the third-generation TKIs completely inhibited metabolic activity of PC9, PC9ER, and H1975 cells (Figs 1B and EV1B). In contrast, 3×mAbs achieved only partial (< 50%) inhibition of metabolic activity, even at relatively high concentrations. Unlike erlotinib, which exerted no consistent effect on EGFR phosphorylation, both third-generation inhibitors we tested, osimertinib and CO-1686 (Sequist *et al*, 2015), strongly reduced EGFR phosphorylation in PC9ER cells, and this was associated with parallel decreases in downstream phosphorylation of AKT and ERK (Fig 1C). Interestingly, 3×mAbs only partly reduced phospho-EGFR signals, an effect attributable to EGFR degradation. Accordingly, the antibodies strongly reduced pERK. However, 3×mAbs spared the basally active AKT. In contrast to the antibodies, the TKIs caused no detectable EGFR degradation, but both inhibitors elevated HER3 (and HER2) surface expression, while 3×mAbs induced down-regulation of surface EGFR, HER2, and HER3 (Fig EV1C), consistent with the reported effect on receptor endocytosis (Mancini *et al*, 2015). In conclusion, the set of *in vitro* assays uncovered remarkable differences between 3×mAbs and osimertinib: While the former reduced surface expression of the target receptors and inhibited pERK, it only partly inhibited metabolism and did not significantly affect pAKT. In contrast, the irreversible TKI strongly inhibited pEGFR, pAKT, pERK, and cellular metabolism, but it up-regulated surface HER3 and HER2.

Next, we compared the ability of 3×mAbs and osimertinib to inhibit tumor growth in mice. Interestingly, both treatments effectively inhibited tumorigenic growth of H1975 cells, but osimertinib achieved an earlier effect (Fig 1D). As expected, both osimertinib and 3×mAbs strongly reduced expression of KI67, a proliferation antigen (Figs 1E and EV1D). The inhibitory effects were reflected also by another test, which administered the two drugs to animals already bearing relatively large H1975 tumors (Fig 1F and G). Immunohistochemical analyses of excised tumors confirmed, on the one hand, the ability of osimertinib to strongly inhibit EGFR phosphorylation and, on the other hand, the ability of 3×mAbs to downregulate EGFR abundance in tumors (Fig EV1E). To address potential toxicities, we analyzed body weights. While animals treated with 3×mAbs gained weight in the course of the experiment (45 days), mice treated with osimertinib displayed slower rates of weight gain (Fig EV1F). In

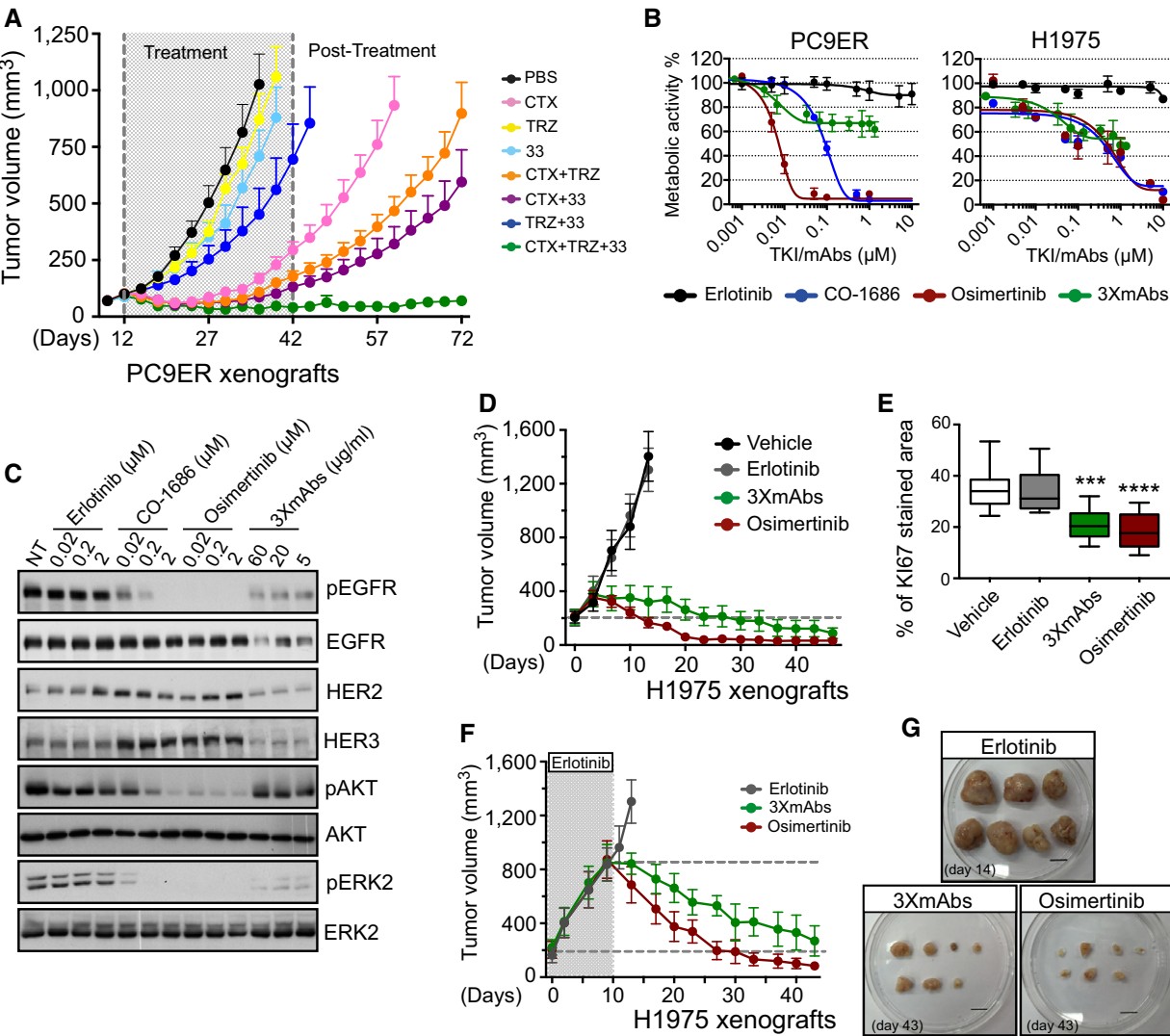

**Figure 1.  Both third-generation TKIs and a triple mAb mixture inhibit erlotinib-resistant tumors, but their mechanisms of action might differ.**

A   PC9ER cells ($4 \times 10^6$ cells per animal) were subcutaneously implanted in CD1-nu/nu mice. Thereafter, tumor-bearing mice were randomized into groups of 9–10 animals that were later treated with the indicated antibodies (0.2 mg/mouse/injection) once every 3 days, for 30 days. Thereafter, tumor growth was followed without any further treatment. Data are means ± SEM from nine mice in each group. CTX, cetuximab; TRZ, trastuzumab; 33, a monoclonal anti-human HER3 antibody; PBS, saline control.

B   Metabolic activity of PC9ER and H1975 cells cultured for 4 days in the presence of increasing concentrations of erlotinib, osimertinib, CO-1686 (rociletinib), or the triple antibody combination (CTX, TRZ, and mAb33). Data are means ± SD values from three experiments.

C   PC9ER cells were treated overnight with the indicated TKIs, or with 3×mAbs, and whole-cell extracts were prepared. Cleared extracts were electrophoresed, and resolved proteins were transferred onto filters. Filters were immunoblotted for the indicated proteins or for their phosphorylated forms. Blots are representative of two independent experiments.

D   H1975 NSCLC cells ($3 \times 10^6$ cells per animal) were subcutaneously grafted in the flanks of CD1-nu/nu mice. Animals were randomized into groups of eight mice after tumors became palpable. Erlotinib (50 mg/kg/dose) and osimertinib (5 mg/kg/dose) were daily administered using oral gavage, whereas the triple antibody combination (3×mAbs; CTX, TRZ, and mAb33; 0.2 mg/mouse/injection) and saline (vehicle) were administered intraperitoneally once every 3 days. Data are means ± SEM values. The broken horizontal line marks the initial tumor volume.

E   H1975 NSCLC cells ($3 \times 10^6$ cells per animal) were subcutaneously grafted in the flanks of four groups of CD1-nu/nu mice. Animals were subjected to the following treatments: erlotinib (50 mg/kg/a daily treatment), osimertinib (5 mg/kg/a daily treatment), or 3×mAbs (CTX, TRZ and mAb33; 0.2 mg/mouse/injection) administered twice a week. Immunohistochemical staining for KI67 in paraffin-embedded sections was performed (see Fig EV1D), and the results are presented in box and whisker plots, where the ends of the box are the upper and lower quartiles, the median is marked by a line inside the box and the whiskers mark the highest and lowest values. Depicting quantifications of KI67 staining using 6–8 sections/tumor. Statistical calculations refer to the control group as reference. ***$P < 0.001$; ****$P < 0.0001$; $n = 5$; one-way ANOVA with Tukey's test.

F, G   H1975 NSCLC cells ($3 \times 10^6$ cells per animal) were subcutaneously grafted in the flanks of three groups of CD1-nu/nu mice. Animals were subjected to erlotinib treatment (50 mg/kg/dose), which continued until tumors reached 800 mm³. Thereafter, each group received one of the following treatments: erlotinib (50 mg/kg/dose), osimertinib (5 mg/kg/dose) or 3×mAbs (CTX, TRZ and mAb33; 0.2 mg/mouse/injection) administered as in (D). Data are means ± SEM from seven mice in each group. Also shown are tumors harvested from each group of animals. Scale bar, 1 cm.

Source data are available online for this figure.

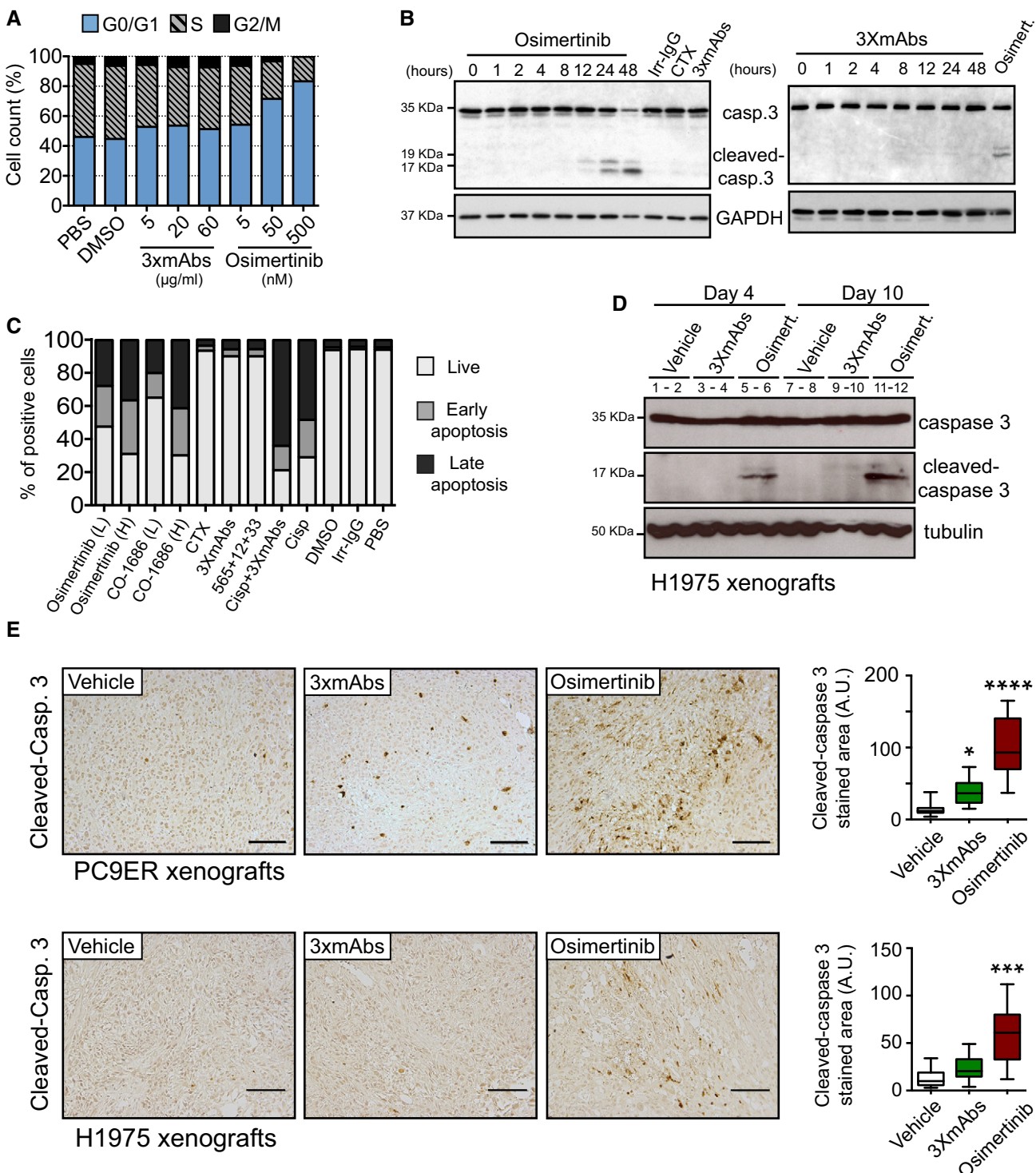

**Figure 2.**

addition, only small differences in favor of fat accumulation in antibody-treated animals were observed when using fat/lean analyses (Fig EV1G). In summary, treatments using osimertinib and 3×mAbs are comparably effective and safe when tested in mice, but the TKI achieves faster kinetics, probably due to complete inhibition of the AKT survival pathway.

### Third-generation TKIs strongly induce apoptosis of erlotinib-resistant cells

In line with a TKI-specific effect on cell growth and survival, we observed a decrease in S-phase cells and a parallel increase in the fraction of cells found in the G0/G1 phase of the cell cycle (Fig 2A).

**Figure 2.  Unlike 3×mAbs, osimertinib induces apoptosis of erlotinib-resistant NSCLC cells.**

A  PC9ER cells were treated for 24 h with increasing concentrations of 3×mAbs or osimertinib, or with the respective vehicles (saline or DMSO). Following incubation with BrdU (60 min), cells were fixed and subjected to BrdU and PI staining. Shown are cell cycle distributions of one representative experiment that used cytometry and 100,000 cells/sample. The experiment was repeated three times.

B  PC9ER cells were treated for the indicated time intervals with osimertinib (0.5 μM) or 3×mAbs (TRZ, CTX, and mAb 33; each at 20 μg/ml). Alternatively, cells were treated for 48 h with an irrelevant immunoglobulin G (Irr-IgG), cetuximab (CTX, 20 μg/ml), osimertinib, or 3×mAbs. Cell extracts were prepared, electrophoresed, and immunoblotted for caspase-3 and its cleaved form. The locations of caspase-3 and two cleaved forms are indicated. GAPDH was used as an equal loading control. Blots are representative of two experiments.

C  PC9ER cells were treated for 48 h with the following agents: saline (PBS), cetuximab (CTX, 20 μg/ml), 3×mAbs (CTX, TRZ, and mAb33, each at 20 μg/ml), osimertinib, CO-1686, cisplatin (1 μM), and a mixture of 3×mAbs and cisplatin. Shown are results of an apoptosis assay performed using an annexin V/7-AAD kit (BioLegend, Inc.). Quantification of the fractions of early and late apoptotic cells is shown (see Fig EV2B). L, low drug concentration (0.01 μM); H, high drug concentration (0.5 μM). The experiment was repeated three times.

D  Mice bearing H1975 tumor xenografts were treated for the indicated time interval with either vehicle, 3×mAbs (0.2 mg/mouse/dose), or with osimertinib (5 mg/kg/dose). Whole tumor extracts were immunoblotted for caspase-3. Note that each lane represents a single tumor.

E  Immunohistochemical staining for cleaved caspase-3 performed on paraffin-embedded sections derived from xenografts of either PC9ER or H1975 cells. Two weeks after tumor inoculation, mice were randomized (3–4 mice/group) and treated for 12 days either with vehicle, 3×mAbs (CTX, TRZ, and mAb33; 0.2 mg/mouse/injection, once every 3 days), or osimertinib (5 mg/kg/injection, once daily). Shown is immunohistochemical staining for the cleaved form of caspase-3 in paraffin-embedded sections. Scale bars, 100 μm. Also shown is a box and whisker analysis (note that the ends of the box are the upper and lower quartiles and the median is marked by a line inside the box) of the results obtained in three different experiments. ****$P < 0.0001$; ***$P < 0.001$; *$P < 0.05$; $n = 3$; one-way ANOVA, with Tukey's test.

Source data are available online for this figure.

Moreover, prolonged incubation of PC9ER cells with osimertinib-induced caspase-3 cleavage, a hallmark of cells undergoing programmed death, but treatment with 3×mAbs was associated with very weak caspase cleavage (Fig 2B). Additional experiments, which are presented in Fig EV2A, employed another marker of apoptosis, namely BIM, which is essential for the action of EGFR kinase inhibitors (Gong *et al*, 2007). The results obtained further supported our conclusion that osimertinib induces stronger cell death signals than the very weak apoptosis effect observed after treatment with 3×mAbs. Probing osimertinib-treated cells for another apoptosis marker, namely annexin V, further supported the notion that the TKI more strongly induced cell death than the antibodies (Figs EV2B and 2C). To help distinguish between early and late apoptosis, we used 7-amino-actinomycin D (7-AAD). Early apoptotic cells exclude 7-AAD, while late-stage apoptotic cells stain positively, due to passage of the dye into the nucleus. Interestingly, whereas a cytotoxic agent, cisplatin, increased primarily late apoptosis, osimertinib increased both early and late apoptosis, and combining cisplatin and 3×mAbs increased late apoptosis (Fig 2C),

in line with a recent study (Ellebaek *et al*, 2016). Consistent with the *in vitro* observations, widespread caspase-3 cleavage was observed in H1975 and in PC9ER xenografts already 4 days after osimertinib treatment (Fig 2D and E). In summary, the third-generation TKI, more than 3×mAbs, induces apoptosis of erlotinib-resistant cells both *in vitro* and in animals.

**The triple antibody combination induces growth arrest and senescence of TKI-resistant NSCLC cells**

Because of the relatively weak apoptosis observed following treatment with 3×mAbs, we considered alternative growth inhibitory mechanisms. Prolonged exposure of PC9ER cells to 3×mAbs induced marked alterations, such as an enlarged and flat morphology (Fig 3A), which might herald cellular senescence (Campisi, 2012; Burton & Krizhanovsky, 2014; Salama *et al*, 2014). Indeed, treatment of cultured PC9 and H1975 cells with 3×mAbs induced prominent activity of SA-β-Gal (senescence-associated β-galactosidase; Figs 3B and EV3A), a hallmark of

**Figure 3.  The triple antibody combination induces senescence of NSCLC cells.**

A  Phase images of PC9ER cells pre-treated for 6 days with either 3×mAbs (20 μg/ml) or saline (CTRL). Osimertinib (0.05 μM) was administrated for 2 days. The internal larger squares represent magnifications of the smaller squares. Scale bar, 100 μm.

B  β-Galactosidase (β-Gal) staining of PC9 cells pre-treated for 11 days with 3×mAbs or saline (CTRL). The internal larger square represents magnifications of the smaller square. Scale bar, 200 μm.

C  Immunoblots of whole extracts isolated from PC9ER cells that were pre-exposed for the indicated time intervals to 3×mAbs (CTX, TRZ, and mAb33). Individual antigens and the respective molecular markers are indicated. GAPDH served as a loading control. Blots are representative of three experiments.

D  β-Gal staining of PC9ER tumor xenografts pre-treated for 14 days with vehicle, 3×mAbs (CTX, TRZ, and mAb33; 0.2 mg/mouse/dose every 3 days), or osimertinib (5 mg/kg/dose every day). Scale bar: 200 μm. The bar plot depicts the respective quantifications (mean ± SD), ****$P < 0.0001$; $n = 4$; one-way ANOVA, with Tukey's test.

E–G  PC9ER cells were treated for 5 days with either 3×mAbs (20 μg/ml), saline (CTRL), or with a low dose of osimertinib (2.5 nM). Shown are representative images of cells fixed in paraformaldehyde and immunostained for KI67 (panel E), γH2AX (panel F), and p16 (panel G). DAPI counterstaining was used to follow nuclear localization. Scale bar, 20 μm. The box and whisker plots, where the ends of the box are the upper and lower quartiles, depict quantifications of results obtained in three independent experiments. **$P ≤ 0.01$, ***$P ≤ 0.001$, $n = 4$; one-way ANOVA with Tukey's test.

H  PC9ER cells ($3 × 10^6$ cells per animal) were implanted in the flanks of CD1-nu/nu mice. Ten days after inoculation, mice were randomized into groups of 3–4 mice and treated for 14 days either with vehicle, 3×mAbs (CTX, TRZ, and mAb33; 0.2 mg/mouse/injection, once every 3 days) or with osimertinib (5 mg/kg/dose, once daily). Shown are tumor sections stained with p16- or γH2AX-specific antibodies. Scale bars, 100 μm. Also shown are bar plots quantifying antigen positivity (mean ± SD). **$P ≤ 0.01$; ***$P < 0.001$; ****$P < 0.0001$; $n = 5$; one-way ANOVA with Tukey's test.

Source data are available online for this figure.

cellular senescence (Dimri *et al*, 1995). Elevation of additional senescence markers was noted *in vitro*, such as DCR2 (TNFRSF10D) and several inhibitors of cyclin-dependent kinases,

namely p16, p21, and p27/KIP (Figs 3C and EV3B). Importantly, analyses of an erlotinib-resistant animal model confirmed that 3×mAbs, more than osimertinib, elevated SA-β-Gal activity

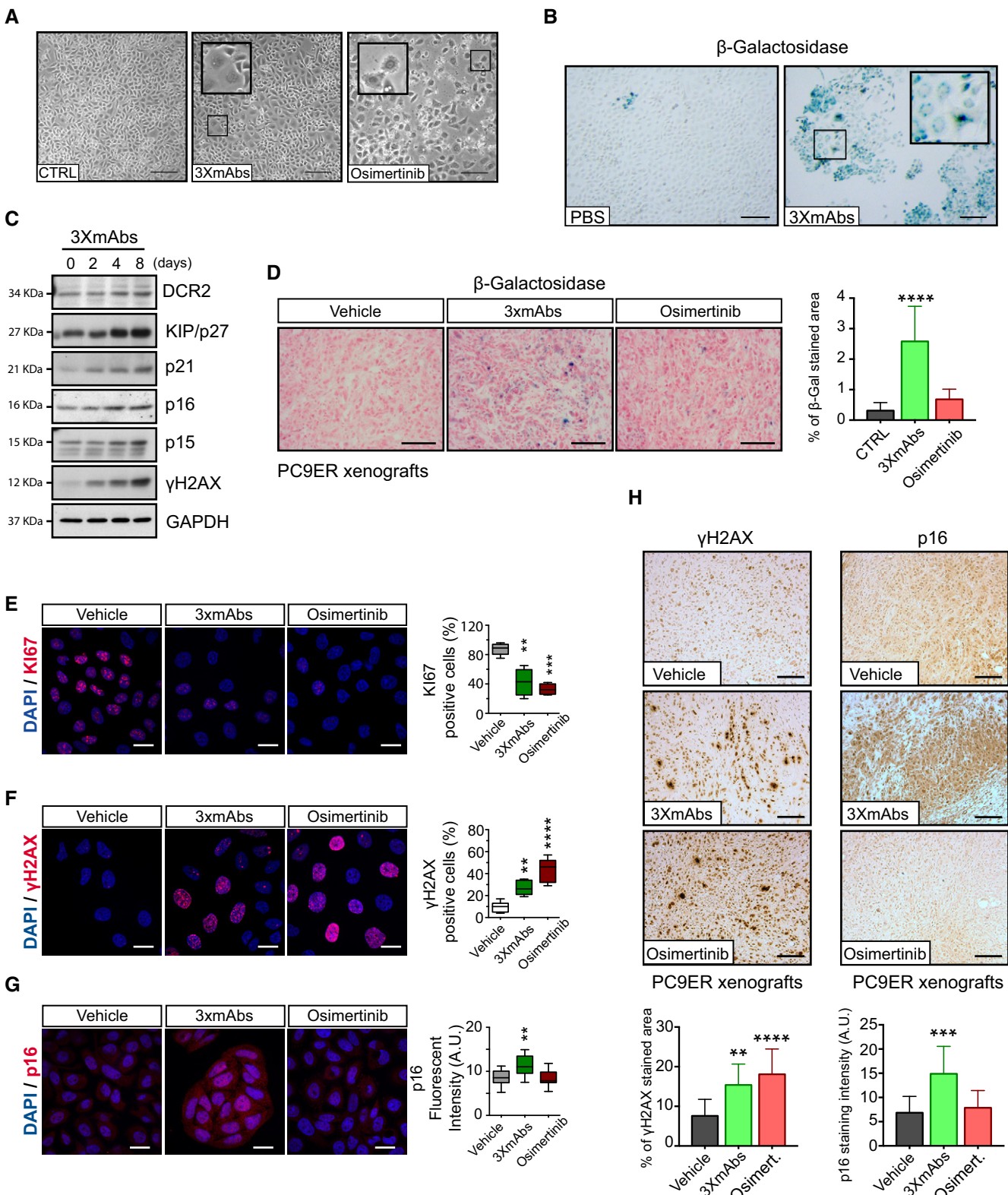

**Figure 3.**

(Fig 3D). As expected, when tested *in vitro*, both treatments reduced KI67 (Fig 3E) and, interestingly, elevated γH2AX (Fig 3F), a component of the DNA damage response (di Fagagna, 2008). However, in line with different modes of action, only 3×mAbs elevated p16 (Fig 3G). Using animal models, we confirmed on the one hand selective effects of 3×mAbs on p16 and p21 and, on the other hand, shared induction of γH2AX by both osimertinib and 3×mAbs (Figs 3H, and EV3C and D). Further *in vitro* tests proposed that the effects of antibodies on γH2AX and another marker of senescence, DCR2, were specifically induced when triple antibody mixtures were tested (Fig EV3E). In summary, antibodies and kinase inhibitors harness different mechanisms of action: 3×mAbs likely acts by means of inducing senescence, whereas osimertinib activates the cell death machinery.

### *In vitro*, 3×mAbs increases ADCC, and *in vivo* the antibody mixture enhances both senescence and recruitment of hematopoietic cells to tumors

Despite mechanistic differences, both osimertinib and 3×mAbs caused rapid shrinkage of pre-established, erlotinib-resistant tumors (Fig 1F). Of relevance, previous studies proposed that secretion of cytokines, including interleukin 8 (IL-8), by cells undergoing senescence enables recruitment of innate immune cells, such as macrophages (Lujambio *et al*, 2013) and natural killers (NK) cells, which can shrink and eliminate tumors (Xue *et al*, 2007; Serrano, 2011; Iannello *et al*, 2013; Marcus *et al*, 2014). Consistent with this scenario, an 8-day-long 3×mAbs treatment of PC9ER cells was followed by increased transcription of *ULBP2* (encoding a ligand for NK cell-specific receptors), while treatment of H1975 cells was associated with up-regulation of both ULBP1 and IL-8, but MICA displayed weaker alterations (Fig EV3F). In this context, it is worthwhile noting that blocking EGFR using a kinase inhibitor was associated with cutaneous inflammatory responses, as well as recruitment of both macrophages and T cells (Lichtenberger *et al*, 2013). In line with these observations, we noted massive recruitment of small, putatively hematopoietic cells to 3×mAbs- and osimertinib-treated H1975 xenografts following drug administration, but cell recruitment was weaker after treatment with erlotinib (Fig EV3G). In addition, when H1975 and NK cells were co-incubated in the presence of 3×mAbs, we observed extensive antibody-dependent cellular cytotoxicity (ADCC; Fig EV3H; note that osimertinib-induced rapid apoptosis precluded determination of TKI-mediated ADCC). In line with these *in vitro* observations, we confirmed *in vivo* that 3×mAbs, unlike osimertinib, can induce beta-galactosidase (β-Gal) activity in H1975 xenografts (Fig EV3I). Taken together, these observations raised the possibility that decoration of senescent tumor cells by mAbs recognizing three different surface proteins may attract lymphocytes and/or myeloid cells, and later instigates immune responses.

### Unlike the irreversible kinase inhibitor, the oligoclonal antibody prevents post-treatment relapses, especially when combined with low-dose osimertinib

In view of the remarkably different mechanisms of action of 3×mAbs and osimertinib, we assumed that their combination might

elicit strong and durable anti-tumor effects. To examine this prediction in an animal model, we adopted a three-step scenario: First, PC9ER cells were implanted in the flanks of CD1-nu/nu mice that were later subjected to erlotinib treatment. On day 20, tumor-bearing mice were randomized into the following arms: (1) vehicle control, (2) erlotinib, (3 and 4) high and low doses of osimertinib, (5) the triple antibody combination (3×mAbs), and (6 and 7) two different combinations of 3×mAbs and osimertinib. These studies confirmed the inhibitory effect of 3×mAbs and the more rapidly evolving effect of osimertinib (Fig 4A and B). Moreover, combining a weakly active dose of osimertinib with 3×mAbs synergistically inhibited tumorigenic growth, while inducing no overt toxicity (Fig EV4A and B). Notably, the combination of 3×mAbs and a TKI achieved complete tumor inhibition, and no relapses were observed in this arm long after treatment termination (> 40 days). In contrast, once osimertinib monotherapy was stopped, tumors rapidly relapsed, but only in one of the nine mice treated with 3×mAbs monotherapy did tumors re-appear, and no relapses were observed when 3×mAbs was co-administered with low- or high-dose oral osimertinib (Figs 4A and B, and EV4C). Immunostaining of the corresponding xenografts for KI67 confirmed that the combination of low-dose osimertinib and 3×mAbs arrested cell proliferation more effectively than each drug alone (Fig 4C and D). In summary, the observations we made in tumor-bearing animals identified an especially effective combination of antibodies and osimertinib that exerts persistent tumor-inhibitory effects, which might reflect the remarkably different mechanisms of action of oligoclonal antibodies and irreversible TKIs.

### Enhancement of receptor endocytosis, as well as more robust apoptosis, may explain the cooperative anti-tumor actions of 3×mAbs and osimertinib

Gratifyingly, we detected the resistance-conferring C797S mutation (Thress *et al*, 2015) in one of six mice that rapidly relapsed post-treatment with high-dose osimertinib (Fig 4E). This observation, together with the augmented anti-proliferation effect of the antibody–TKI combination (Fig 4D), prompted us to investigate mechanism(s) underlying inhibition of relapse of mutation-bearing and possibly other resistant clones. We previously attributed the anti-tumor effects of 3×mAbs to simultaneous endocytosis and degradation of the three target receptors for EGF-like growth factors (Mancini *et al*, 2015). In line with this interpretation, the combination of two drugs more effectively destabilized HER3, HER2, and EGFR in PC9ER cells (Fig 5A). Furthermore, using cytometry, this effect was linked to removal of the three receptors from the cell surface (Fig 5B). Interestingly, we noted that *in vitro* treatment with osimertinib augmented surface expression, along with overall abundance of HER3 and HER2 (Fig 5B), suggesting that TKI monotherapy might enhance recognition of mutant-expressing cells by 3×mAbs.

To examine *in vivo* relevance of antibody-induced endocytosis and degradation, we treated mice harboring H1975 xenografts for 10 days. Thereafter, tumors were harvested and fixed and thin slices were analyzed by immunofluorescence using anti-EGFR, anti-HER2, and anti-HER3 antibodies (Fig 5C). As expected, we observed bright staining with all three antibodies when untreated, or when erlotinib-treated tumors were examined. Treatment with osimertinib

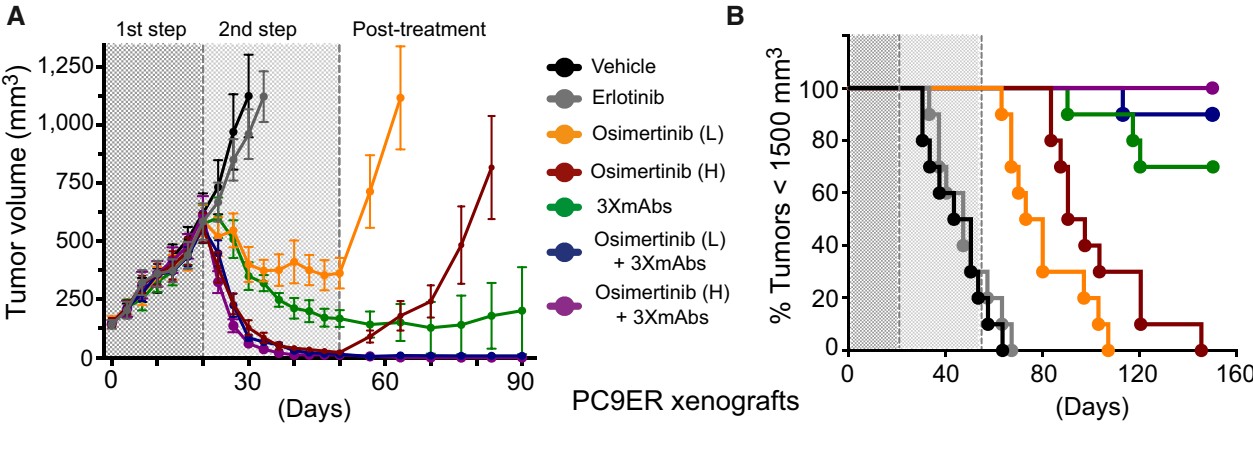

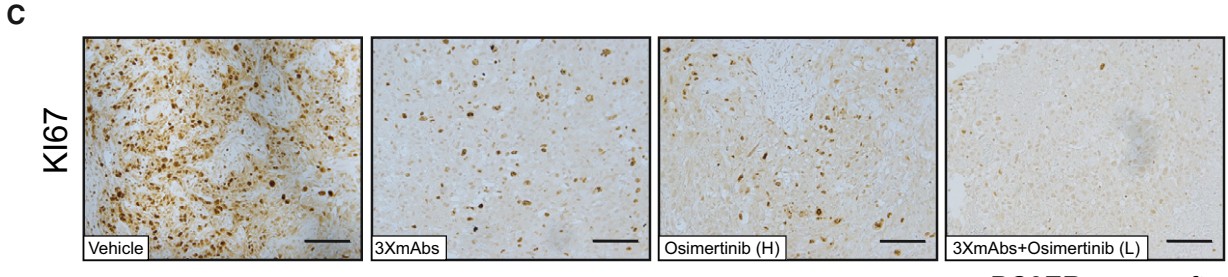

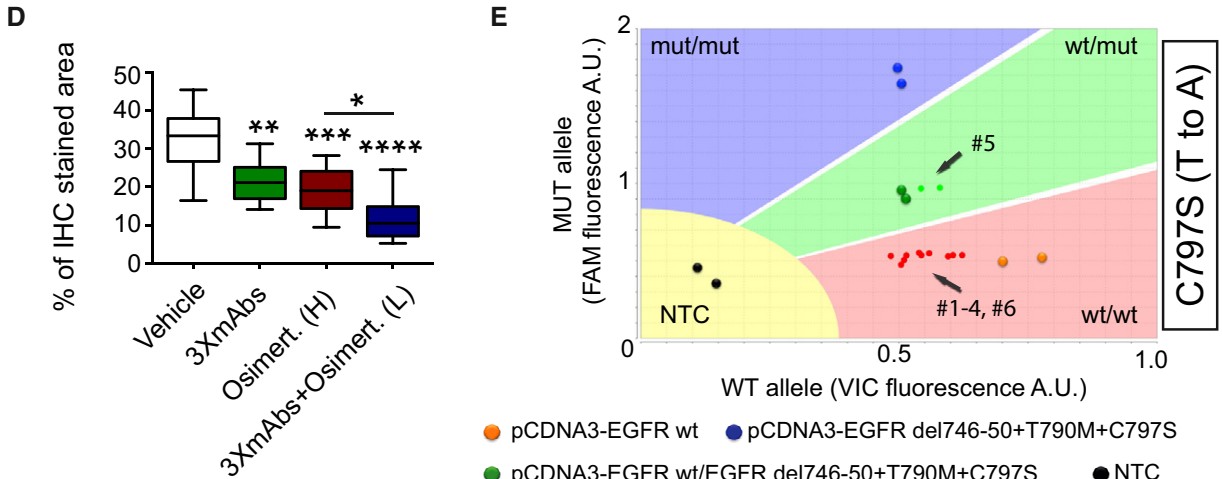

**Figure 4.  Unlike the kinase inhibitor, the oligoclonal antibody prevents post-treatment relapses, especially when combined with low-dose osimertinib.**

A    PC9ER cells ($3 \times 10^6$ cells per animal) were implanted in the flanks of CD1-nu/nu mice. Once palpable tumors emerged, mice were daily treated with erlotinib (50 mg/kg/dose; 1st step). Following randomization of mice, we started the following arms of second step treatment: (1) vehicle control, (2) erlotinib, (3 and 4) osimertinib (1 or 5 mg/kg/dose; L or H, respectively), (5) the triple antibody combination, which was injected intraperitoneally once every 3 days, (6 and 7) and combinations of 3×mAbs (0.2 mg/mouse/dose) and osimertinib, either high or low. All treatments were stopped on day 50, but we continued monitoring tumor volumes (post-treatment phase). Data are means ± SEM from 9 to 10 mice of each group.

B    Kaplan–Meier survival analysis of the tumor-bearing mice shown in (A). Mice were euthanized when tumor volumes reached 1,500 mm³.

C, D  One week after inoculation of PC9ER cells, mice were randomized (4 mice/group) and treated for 14 days as described in (A) (arms 1, 4, 5, and 7). Shown is immunohistochemical staining for KI67 in paraffin-embedded sections using specific antibodies. Scale bars, 100 μm. Also shown are box and whisker plots (note that the ends of the box are the upper and lower quartiles and the median is marked by a line inside the box) depicting quantifications of KI67 staining. *$P < 0.05$; **$P < 0.01$; ***$P < 0.001$; ****$P < 0.0001$; $n = 6$; one-way ANOVA with Tukey's test.

E    Genomic DNA, along with the indicated plasmid DNA reference samples, were tested for EGFR's T790M and C797S mutations using the TaqMan genotyping assay. DNA was isolated from six relapsing tumors, all from mice treated with osimertinib (high dose; arm 4 in panel A). Fluorescent signals corresponding to individual tumors and mice are shown as small colored dots, whereas large dots represent signals obtained from plasmid DNA. Note that VIC-labeled primers were used for amplification of wild-type alleles, whereas FAM-labeled primers were used for the mutant alleles. As indicated, only one animal, mouse number 5 (#5), scored positive for a mutant C797S allele. Data were analyzed using TaqMan Genotyper Software (Life technologies). NTC, no-template control.

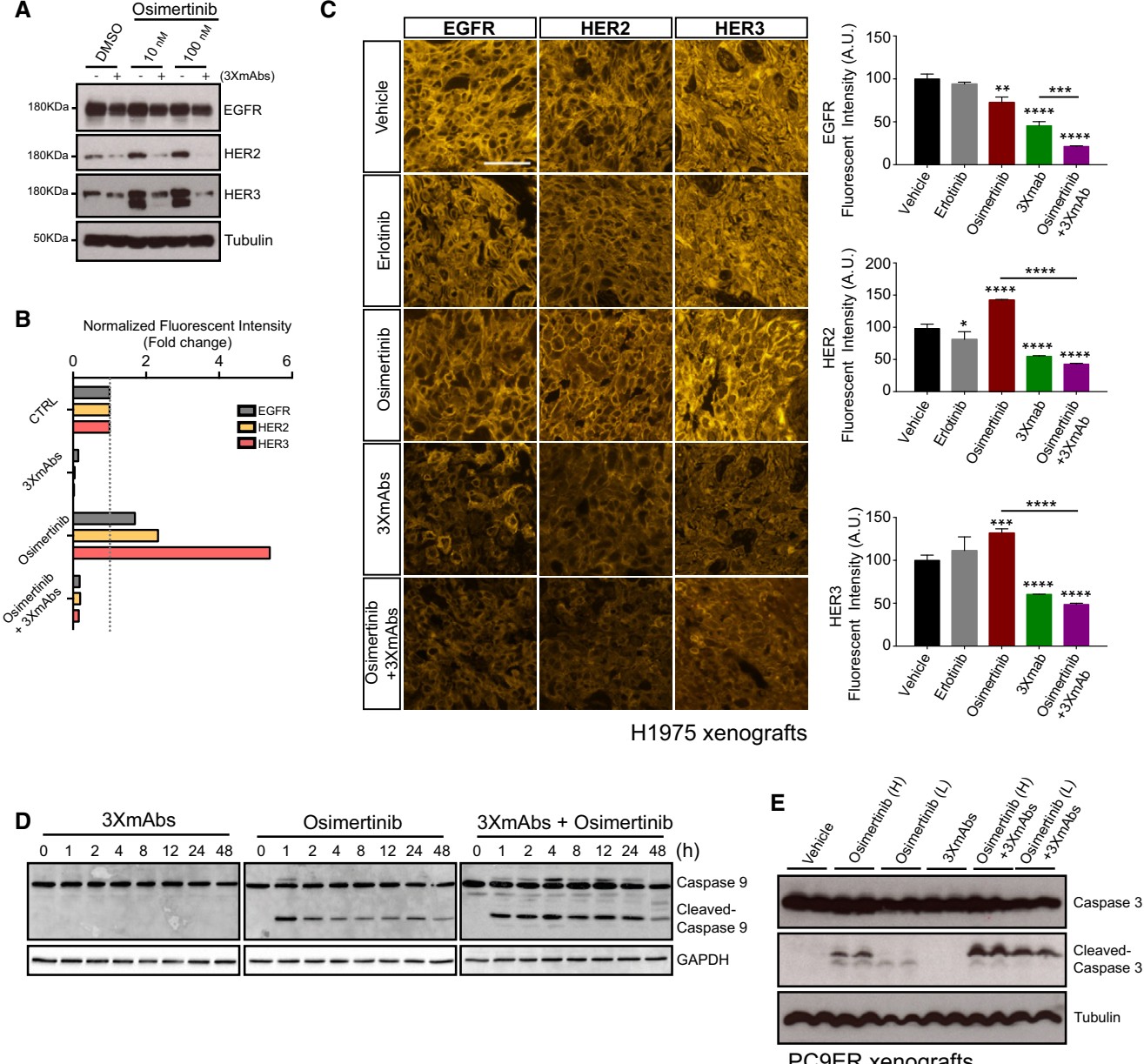

H1975 xenografts

PC9ER xenografts

**Figure 5.  Enhancement of receptor endocytosis, as well as cellular apoptosis, may explain the cooperative anti-tumor actions of 3×mAbs and osimertinib.**

A    PC9ER cells were treated for 24 h with DMSO (control) or osimertinib (10 nM or 100 nM), in the absence or presence of 3×mAbs (30 μg/ml). Whole-cell extracts were analyzed using electrophoresis and immunoblotting. Blots are representative of three independent experiments.

B    PC9ER cells were treated for 24 h with saline (CTRL), 3×mAbs (40 μg/ml), osimertinib (500 nM), or with the combination (3×mAbs plus osimertinib). Thereafter, cells were analyzed using flow cytometry for surface expression levels of EGFR, HER2, and HER3. Normalized surface expression levels are presented as means ± SEM of two independent experiments.

C    CD1-nu/nu mice harboring H1975 xenografts were treated for 10 days, as indicated. Thereafter, tumors were harvested, formalin-fixed, and paraffin-embedded (FFPE). The corresponding sections (4 μm), obtained from each tissue block were analyzed by immunofluorescence using anti-EGFR, anti-HER2, or anti-HER3 antibodies. Afterward, sections were washed and incubated for 90 min at room temperature with a biotinylated anti-rabbit secondary antibody, followed by Cy3-conjugated streptavidin. Sections were washed and covered with Aqua Polymount. Stained sections were examined and photographed using a fluorescence microscope (Eclipse Ni-U, Nikon, Tokyo, Japan) equipped with a Plan Fluor objectives (20×) connected to a monochrome camera (DS-Qi1, Nikon). Scale bar, 25 μm. Also shown are histograms depicting quantifications of receptor staining (mean ± SD). *P < 0.05; **P < 0.01; ***P < 0.001; ****P < 0.0001; n = 3; one-way ANOVA with Tukey's test.

D    PC9ER cells were treated for the indicated time intervals with osimertinib (500 nM), 3×mAbs (20 μg/ml), or the combination of both treatments. Whole-cell extracts were analyzed for caspase-9 and cleavage products. GAPDH levels were used for comparison of gel loading.

E    Mice bearing PC9ER tumor xenografts were treated for 7 days with either vehicle, 3×mAbs (0.2 mg/mouse/dose), osimertinib high (H; 5 mg/kg/dose), or low (L; 1 mg/kg/ dose), or with the indicated combinations. Tumor extracts were probed for caspase-3 and the cleaved form. Tubulin was used to compare the amounts of protein loaded.

Source data are available online for this figure.

was associated with the appearance of some necrotic foci, along with moderate effects on receptor levels. As predicted by our *in vitro* assays, significantly stronger negative effects on the abundance of the three receptors were observed when tumors were pre-treated with 3×mAbs, either alone or in combination with osimertinib.

To address apoptosis, we utilized three different molecular markers and inferred enhancement of TKI-induced apoptosis by 3×mAbs. Thus, 3×mAbs instigated weak cleavage of PARP (poly(ADP-ribose) polymerase), a marker of apoptosis, but the signals observed after treatment with the drug combination (especially low doses of osimertinib) exceeded those induced by each agent alone (Fig EV4D). Likewise, augmented osimertinib-induced cleavage of caspase-3 was observed following a prolonged *in vitro* treatment (11 days) of PC9ER cells with 3×mAbs, although when applied alone the antibodies induced no cleavage (Fig EV4E). Consistent with caspase-3 activation, analysis of an upstream protease, caspase-9, confirmed that the combination of drugs enhanced appearance of the p17 cleaved form (Fig 5D). In a similar manner, although osimertinib alone induced activation of BIM and 3×mAbs was weakly effective, enhanced BIM activation was observed following treatment with the combination (Fig EV4F). For *in vivo* relevance, we treated mice bearing PC9ER xenografts with osimertinib, 3×mAbs, or the combinations and observed, 7 days later, a clear enhancement of caspase-3 cleavage (Fig 5E), similar to the results obtained using cultured cells.

Next, we addressed the prediction that antibody-induced enhancement of apoptosis translates to stronger inhibition of cell proliferation. For example, 3×mAbs induced only a small effect on the S-phase fraction of PC9ER cells, but cell division almost stopped when the antibodies were combined with osimertinib (Fig EV4G), which might explain the synergistic action of the antibody plus TKI combination. Taken together, the cooperative effects of the combination of osimertinib and three antibodies, in terms of both endocytosis and apoptosis, are consistent with the pharmacological prediction of collaboration between two drugs employing distinct modes of action.

## Cells that acquire the C797S mutation, which confers resistance to osimertinib, remain sensitive to the triple antibody combination

Because spontaneous emergence of the C797S mutation was observed in an animal model, as shown in Fig 4E, we were prompted to establish a parallel *in vitro* system of acquired resistance. To this end, we incubated PC9ER cells for 4 months with osimertinib and isolated surviving cells (hereinafter, PC9ER-AZDR cells). As expected, the IC50 value of PC9ER-AZDR cells toward the inhibitor increased by 2–3 orders of magnitude, relative to the parental cells (Figs 6A and EV5A), the drug lost the ability to inhibit phosphorylation of EGFR, AKT, and ERK (Fig 6B), and initial genotyping proposed emergence of the C797S mutation (Fig EV5B). Interestingly, PC9ER-AZDR cells also acquired resistance to CO-1686, another third-generation TKI. One potentially shared mechanism of resistance might involve elevated EGFR abundance, as revealed by immunoblot and real-time PCR analyses (Figs 6B and C, and EV5C). Because PC9ER-AZDR cells also acquired resistance to afatinib (Fig EV5A), an irreversible TKI that depends on an intact cysteine at position 797, we analyzed in greater depth the emergence of the previously reported C797S mutation (Thress *et al*, 2015). Indeed, by applying a digital PCR-based assay on genomic DNA extracted from PC9ER-AZDR cells, we confirmed the presence of the C797S mutation (Fig EV5B–D; see Appendix Table S1). It is notable that we mapped the two resistance-promoting mutations, namely T790M and C797S, to the same allele, a configuration preempting combined treatment with first- and third-generation EGFR inhibitors. Moreover, determination of copy number variation using digital PCR and the Fluidigm Biomark platform indicated that acquisition of osimertinib resistance involved amplification of the del746/750 allele, along with selective enrichment for the

---

**Figure 6.   The oligoclonal antibody overcomes the C797S mutation-mediated resistance to osimertinib.**

A   Survival assays of PC9ER and derivative PC9ER-AZDR cells, which were selected *in vitro* for resistance to osimertinib. Cells were treated for 72 h with increasing concentrations of osimertinib, or with the triple combination of antibodies (3×mAbs; cetuximab, trastuzumab, and mAb33). Data are means ± SD values of three independent experiments.

B   Immunoblotting of PC9ER and PC9ER-AZDR cells treated for 6 h with DMSO (vehicle control), osimertinib (1 μM), or CO-1686 (1 μM). Whole-cell extracts were analyzed using electrophoresis and immunoblotting. Blots are representative of two independent experiments.

C   Total RNA was isolated from PC9ER and PC9ER-AZDR cells, complementary DNA was synthesized, and specific *EGFR* primers were employed. Shown are results of real-time qPCR analysis after normalization to *GAPDH*. The assay was repeated three times. Signals represent means ± SD values; **$P < 0.05$; $n = 3$; $t$-test.

D   Copy number variation was determined using digital PCR. Shown are the states of both wild-type (WT) and EGFR mutations (mut; del746/750, T790M or C797S) in DNA extracted from PC9ER and PC9ER-AZDR cells. RNASEP was used as normalizer. Error bars indicate the Poisson 95% confidence intervals for each copy number determination.

E   PC9ER-AZDR cells ($3 \times 10^6$ cells per animal) were subcutaneously grafted in CD1-nu/nu mice. Thereafter, tumor-bearing animals were randomized into groups of 9–10 mice, which were later treated once every 3 days with 3×mAbs (0.2 mg/mouse/injection). Alternatively, mice were orally treated with the following TKIs (or vehicle): erlotinib (50 mg/kg/dose) or two doses of osimertinib (1 or 5 mg/kg/dose; L or H, respectively). Tumor growth was monitored once every 3 days. Averages ± SEM values are presented.

F   PC9ER-AZDR cells ($3 \times 10^6$ cells per animal) were subcutaneously grafted in CD1-nu/nu mice. Thereafter, tumor-bearing animals were randomized into groups of 9–10 mice, which were later treated once every 3 days with 3×mAbs (0.2 mg/mouse/injection). Alternatively, mice were orally treated with two doses of osimertinib (1 or 5 mg/kg/dose; L or H, respectively). Shown is immunohistochemical staining of paraffin-embedded sections for KI67. Scale bars, 100 μm. Also shown are box and whisker plots depicting quantifications of KI67 staining using 6–8 sections/tumor. ***$P < 0.001$; $n = 4$; one-way ANOVA with Tukey's test.

G   β-Galactosidase (β-Gal) staining of PC9ER-AZDR tumor xenografts. Tumor-bearing mice were treated for 20 days with either 3×mAbs (CTX, TRZ, and mAb33; 0.2 mg/mouse/injection, once every 3 days) or saline. Scale bar, 200 μm.

Source data are available online for this figure.

T790M-containing allele and *de novo* emergence of the C797S allele (Figs 6D, and EV5E and F, and Appendix Table S2).

To address in animals the ability of 3×mAbs to overcome resistance to osimertinib, we inoculated PC9ER-AZDR cells in CD1-nu/nu mice, which were later treated with either osimertinib or antibodies. As expected, neither erlotinib nor osimertinib inhibited growth of the corresponding tumors in animals (Fig 6E). On the contrary, treatment with 3×mAbs strongly and persistently reduced tumor volumes, and no relapses were observed > 90 days after ending all antibody treatments. Immunohistochemical staining of xenografts confirmed the ability of 3×mAbs, unlike osimertinib, to arrest cell proliferation (Fig 6F), as well as increase cellular senescence, as revealed by β-galactosidase activity (Fig 6G), in line with a senescence-associated mechanism of tumor inhibition.

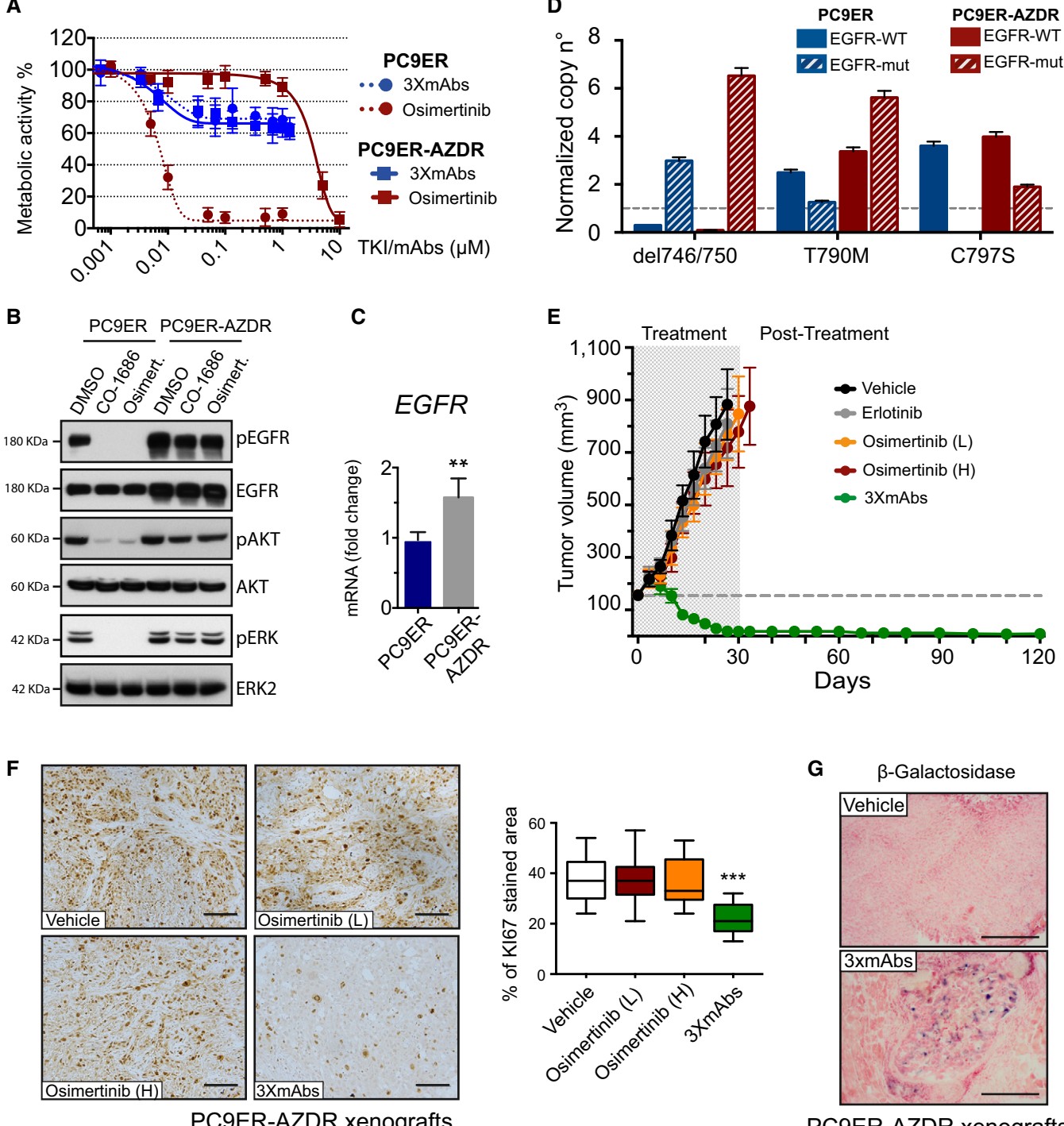

Figure 6.

In conclusion, NSCLC cells that acquired resistance to first-, second-, and third-generation TKIs remain sensitive to an antibody combination able to simultaneously down-regulate EGFR, HER2, and HER3. Hence, it is conceivable that 3×mAbs would be able to nullify, in clinical settings, TKI resistance driven by the C797S mutation. Our future studies will address, on the one hand, the ability of 3×mAbs to overcome other mechanisms of resistance to irreversible TKIs and, on the other hand, mechanisms of putative resistance to 3×mAbs, possibly due to activation of alternative driver oncogenes (e.g., *MET* amplification) or to emergence of downstream mutations (e.g., in *NRAS*) (Eberlein *et al*, 2015).

## Discussion

Osimertinib is the first clinically approved drug that specifically targets T790M-mutated EGFRs (Janne *et al*, 2015), but several mechanisms, including emergence of the C797S mutation, will likely limit durability of responses to the new, third-generation inhibitors (Eberlein *et al*, 2015; Thress *et al*, 2015). The rationale behind developing 3×mAbs as a potential alternative strategy is dual: (i) the combination of three antibodies nullifies compensatory feedback loops involving HER2 and HER3 (Mancini *et al*, 2015), and (ii) 3×mAbs recognizes the genetically more stable parts of EGFR, HER2, and HER3, namely the extracellular domains, independently of the identity and number of intracellular mutations. Our animal studies, although confined to only two model cell lines, indicate that all three mAbs must be co-administered in order to fully eliminate pre-established NSCLC tumors. Although this therapeutic effect develops slowly, in comparison with the rapid effect of osimertinib, in our models only rarely did recurrence slowly follow completion of 3×mAbs treatment. This is in marked contrast to osimertinib: Soon after complete regression and treatment cessation, tumors rapidly regrow. Notably, according to a recent study, relapses may be due to a variety of mechanisms, including pre-existing mutants of EGFR and newly generated mutations (Hata *et al*, 2016; Ramirez *et al*, 2016). Because only one of six animals relapsing after osimertinib treatment tested positive for the C797S mutation, and a parallel experimental arm showed strong inhibition of relapses by 3×mAbs, we assume that 3×mAbs also blocks alternative modes of resistance to osimertinib. Altogether, our data indicate that 3×mAbs is active against the following mutations: L858R, del746-750, T790M, and C797S. This list predicts that yet unknown kinase domain mutations will show similar sensitivities. Nevertheless, co-administration of three different mAbs is expected to cause some adverse effects, such as skin and gastrointestinal tract toxicities (Kol *et al*, 2014; Gaborit *et al*, 2015).

Several mechanisms may underlay the ability of mAbs to inhibit tumor growth. They include recruitment of immune effector cells, complement-dependent cytotoxicity, interception of pathways essential for tumorigenesis, induction of apoptosis, and stimulation of receptor degradation (Carvalho *et al*, 2016). However, so far the contribution of cellular senescence has not been addressed. It is notable that cells undergoing senescence not only arrest their own proliferation; by secreting cytokines, they recruit and activate immune cells, including macrophages and NK cells (Munoz-Espin & Serrano, 2014). Furthermore, while undergoing senescence, cells can up-regulate immune ligands on their own surface, thereby engage mechanisms capable of eliminating senescent cells (Sagiv & Krizhanovsky, 2013). Because our immunocompromised animal model precludes in-depth analysis of immune mechanisms leading to tumor shrinkage, we focused on non-immune processes, such as cellular senescence and antibody-mediated endocytosis and degradation of receptors for survival factors.

It is notable that our animal studies uncovered cooperative interactions between 3×mAbs and low-dose osimertinib: following a relatively short treatment (4 weeks), all treated mice were cured by the mAb-TKI combination. We attribute synergy to, on the one hand, the ability of 3×mAbs to sort survival receptors for degradation, thereby instigate senescence, and, on the other hand, to the ability of osimertinib to strongly induce apoptosis. In other words, the mAb-TKI combination harnesses two different inhibitory mechanisms, which might augment each other. Although we did not address immunological mechanisms, it is worthwhile indicating that trastuzumab can mediate ADCC in animals (Clynes *et al*, 2000) and polymorphic isoforms of Fcγ receptors, which mediate ADCC, have been implicated in responses of colorectal and breast cancer patients to cetuximab (Zhang *et al*, 2007; Bibeau *et al*, 2009) and trastuzumab (Musolino *et al*, 2008), respectively. Along a similar vein, complement-mediated anti-tumor activity of trastuzumab has been demonstrated (Mamidi *et al*, 2013). Examination of such *in vivo* actions of 3×mAbs is currently limited by the availability of suitable animal models.

In summary, our studies identify a combination of three antibodies, two of them are clinically approved for other diseases, as a potent treatment able to persistently inhibit models of EGFR-driven NSCLC. The data imply that the triple antibody mixture will block all kinase domain mutants of EGFR, hence avoid repeated cycles of resistance due to emergence of new mutations. Furthermore, combining the oligoclonal antibody with a low dose of a third-generation inhibitor of EGFR improved anti-tumor efficacy, avoided recurrence, and durably cleared tumors in mice, features that may justify larger scale tests of 3×mAbs, either alone or in combinations, in clinical settings.

## Materials and Methods

### Real-time and digital PCR (dPCR)

Total RNA was isolated using PerfectPure RNA Cultured Cell Kit (5-prime, Hamburg). Complementary DNA was synthesized using the miScript kit (QIAGEN). Primers were designed using Primer3. Real-time qPCR analyses were performed using SYBR Green (Qiagen or Applied Biosystems) and specific primers (see a list of primers in Appendix Table S3). qPCR signals (cT) were normalized to *B2M*. Digital PCR experiments were performed using the Fluidigm Biomark platform in 12.765 chip format. The primers we used are listed in Appendix Table S4. All dPCR reagents and assays were from Life Technologies. Genomic DNA was isolated from cell lines using Pure Link Genomic DNA mini kit. Stocks (10 ng/µl) of each DNA sample were serially diluted (1:1, 1:2, 1:4, 1:8, 1:16). For reference, we used two control plasmid DNAs (1 pg): pCDNA3-EGFRwt and pCDNA3-EGFRdel746/50 + T790M + C797S. For primer sequences, see a list in Appendix Table S3. RNase P, copy number reference assay, and all of the other SNP reagents were obtained

from Applied Biosystems. Cycling conditions were as follows: 60°C for 30 s (1 cycle), 95°C for 10 min (1 cycle), 40 cycles of 94°C for 15 s and 60°C for 1 min, followed by 10°C hold. The count of partitions showing positive amplification was obtained using Fluidigm Digital PCR Analysis software. This count was analyzed after Poisson correction, to account for the fact that some positive partitions might contain more than one DNA molecule. As the number of positive partitions increases, so does the probability that some partitions will contain more than one target molecule (Sanders *et al*, 2013). The corrected values were provided by the Fluidigm software as "Estimated Targets". To calculate the number of copies per ng of genomic DNA, we used the following equation: no. of copy/ngDNA = [no. of Est. Targets/4.59 (µl/panel)] * [10 (µl reaction volume)/1 (µl DNA)]/ 10 ng (DNA/µl)] * dilution factor.

### Genotyping assays

Genomic DNA was extracted from cell lines or from tumor xenografts using the Pure Link genomic DNA Kit (Invitrogen). Three TaqMan SNP genotyping assays (Applied Biosystems, Foster City, CA, USA) were custom designed to detect specific EGFR mutations: del476/750, T790M, and C797S (see a list of primers and probes in Appendix Table S4). Reactions were carried out in the StepOnePlus real-time PCR system, using 10 ng of DNA per sample. Cycling conditions were as follows: 60°C for 30 s (one cycle), 95°C for 10 min (one cycle), 40 cycles at 94°C for 15 s, and 60°C for 1 min, followed by 30 s at 60°C. As a negative control, we used a sample without template (NTC). All samples were run in duplicates, and experiments were repeated three times.

### *In vitro* monitoring cytotoxicity using the real-time cell analyzer (RTCA)

ADCC activity was examined using RTCA. H1975 cells were pretreated for 8 days with saline, CTX, 3×mAbs (20 µg/ml), or for 3 days with osimertinib (0.0025 µM). Thereafter, cells were transferred to RTCA's E-plates (8,000 cells per well) and monitored overnight. Before adding NK cells, the previously indicated treatments were refreshed. NK92 cells were added in a volume of 100 µl per well. Treatment with 1% Triton X-100 was used as an indicator of maximal elimination of target cells.

### Tumorigenic cell growth in mice

All animal studies were approved by the Weizmann Institute's Review Board (IRB). CD1-nu/nu or NSG mice (8–9 weeks old females) were randomized into groups of 6–10 mice. Mice were injected subcutaneously in the right flank with cancer cells ($3 \times 10^6$ or $4 \times 10^6$ per mouse). mAbs were injected intraperitoneally at 200 µg per mouse per injection, once every 3 days. Daily administration of TKIs used oral gavage. Tumor width (W) and length (L) were measured once a week with a caliper, and tumor volume (V) was calculated according to the following formula: $V = 3.14 \times (W^2 \times L)/6$. Body weight was evaluated once per week. Mice were euthanized when tumor size reached 1,500 mm³. Body composition analysis was performed using the Minispec LF50 Body Composition Analyzer (from Bruker). We assessed lean tissue, fat,

### The paper explained

#### Problem

Many anti-cancer drugs specifically inhibit mutant forms of tyrosine kinases, but their application is limited by rapidly evolving tumor adaptations. For example, a fraction of lung tumors is driven by mutations within *EGFR*, which can be inhibited by specific drugs, but emergence of a single secondary mutation, T790M, is the major cause of acquired patient resistance. This may be prevented by the newly approved third-generation kinase inhibitor called osimertinib, but several mechanisms, including a third-site mutation, C797S, confer resistance to osimertinib. Treatments that durably overcome recurring resistance to EGFR inhibitors are urgently needed.

#### Results

To overcome resistance to the first-generation EGFR inhibitors, we previously applied a triple mixture of monoclonal antibodies, 3×mAbs, targeting EGFR, HER2, and HER3. Herein, we compare 3×mAbs and a third-generation inhibitor, osimertinib, and show that simultaneous blockade of EGFR, along with its two siblings, is both necessary and sufficient for preventing evolvement of resistance to the newly approved drug. Our biochemical and animal studies showed that osimertinib's mechanism of action entails promotion of tumor cell death by inhibiting the AKT pathway. This mode of action is remarkably different from the mode of action of the antibody mixture, which accelerates endocytosis of the three receptors, hence arrests cell growth with minimal cell killing. These observations predicted synergy between osimertinib and 3×mAbs, which we verified in animals: A weakly active low dose of osimertinib enabled 3×mAbs to durably and completely eradicate already established drug-resistant tumors. Taken together, our data indicate that targeting the extracellular domain of EGFR and its siblings, by means of a three-antibody mixture, can overcome emergence of resistance to drugs targeting the intracellular kinase domain of EGFR.

#### Impact

Our study offers a potential treatment for many lung cancer patients, who sequentially develop resistance to first-, second- and third-generation EGFR inhibitors. Whether or not the combination of drugs we propose will overcome also additional mechanisms of tolerance to kinase inhibitors remains an open question.

and fluids in live mice. Investigators were not blinded to the group allocation during the experiment and when assessing outcome.

### Statistical and data analyses

Cell sorting data were analyzed using the FlowJo v10.0.7 software (Tree Star, Inc., Ashland, OR, USA). Staining intensity was determined using ImageJ. All data were analyzed using the Prism GraphPad software, and statistical analyses were performed using one or two-way ANOVA with Tukey or Bonferroni's test (*$P \leq 0.05$; **$P \leq 0.01$; ***$P \leq 0.001$; ****$P \leq 0.0001$). For the analysis of real-time PCR data, we applied two-tailed *t*-test (*$P \leq 0.05$; **$P \leq 0.01$). For exact *P*-values and additional statistical information, see Appendix Table S5.

**Expanded View** for this article is available online.

### Acknowledgements

We thank Michael Sela for continuous help and Alon Harmelin, Inbal Biton, Nava Nevo, Ori Brenner, and Naama Feldman for guiding our animal studies.

Our laboratory was supported by the European Research Council, the Israel Science Foundation, the Seventh Framework Program of the European Commission (LungTarget Consortium), the Israel Cancer Research Fund and the Dr. Miriam and Sheldon G. Adelson Medical Research Foundation. Y.Y. is the incumbent of the Harold and Zelda Goldenberg Professorial Chair. Our studies were performed in the Marvin Tanner Laboratory for Research on Cancer.

## Author contributions

MM and YY designed the study; MM, HG, NG, LM, DR, TMS, ML, GM, YE, DGAB, LR, AN, IM, and REA performed the experiments; MM, VK, DA, and YY analyzed data; EB, JD, and AM provided reagents; MM, VK, and YY wrote the manuscript. All authors declare no competing interests. Requests for reagents should be addressed to YY.

## Conflict of interest

Yeda, the technology transfer office of the Weizmann Institute, protected relevant intellectual property. All authors declare that they have no other conflict of interests.

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
