## [Review Process File · EMBO Molecular Medicine]

An Oligoclonal Antibody Durably Overcomes Resistance of Lung Cancer to Third Generation EGFR Inhibitors

Maicol Mancini, Hilah Gal, Nadège Gaborit, Luigi Mazzeo, Donatella Romaniello, Tomer Meir Salame, Moshit Lindzen, Georg Mahlknecht, Yehoshua Enuka, Dominick G. A. Burton, Lee Roth, Ashish Noronha, Ilaria Marrocco, Dan Adreka, Raya Eilam Altstadter, Emilie Bousquet, Julian Downward, Antonio Maraver, Valery Krizhanovsky and Yosef Yarden

Corresponding author: Yosef Yarden, The Weizmann Institute of Science

Review timeline:

Submission date:	26 May 2017
Editorial Decision:	19 June 2017
Revision received:	23 September 2017
Editorial Decision:	10 October 2017
Revision received:	23 October 2017
Accepted:	30 October 2017

Transaction Report:

Editors: Roberto Buccione and Céline Carret

1st Editorial Decision

19 June 2017

Thank you for the submission of your manuscript to EMBO Molecular Medicine. We have now heard back from the three Reviewers whom we asked to evaluate your manuscript

As you will see the reviewers are very positive, although they do express some concerns on your manuscript, which I would basically summarise as follows: 1) the limited scope of the models used, 2) the need to extend the observations on proliferation, apoptosis and senescence to all experimental settings and 3) the unsuitability of the experimental models to support conclusions on the role of the immune system.

After our reviewer cross-commenting exercise, there was full agreement among the reviewers that the above concerns (in addition to the other items) would need to be addressed but there was also a clear consensus that we would not be asking you to address the immune aspect. It was actually suggested that it would be altogether better to remove that part from the manuscript to improve both its focus and clarity.

In conclusion, while publication of the manuscript cannot be considered at this stage, we would be pleased to consider a revised submission, with the understanding that the Reviewers' concerns must be addressed including further experimentation as per the indications mentioned above. Eventual acceptance of the manuscript will entail a second round of review.

I look forward to seeing a revised form of your manuscript in due time.

***** Reviewer's comments *****

Referee #1 (Remarks):

Mancini et al. examined the effect of a triple mixture of monoclonal antibodies targeting EGFR, HER2 and HER3 (3XmAbs), on first-generation EGFR inhibitor (1G EGFRi) resistant lung cancer models and compared their anti-cancer properties with the third-generation (3G) EGFR inhibitor osimertinib. The authors show that osimertinib and 3XmAbs comparably inhibit 1G EGFRi-resistant cell lines in vitro and in vivo. However, while osimertinib induces extensive apoptosis in 1G EGFRi-resistant tumours, 3XmAbs induce senescence. Interestingly, tumour relapse is prevented in tumour-bearing animals treated with a combination of 3XmAbs and osimertinib which is likely due to the distinct mechanisms of action of each drug. Previous studies have shown that patients treated with 3G EGFRi such as osimertinib acquire resistance, mostly due to the emergence of C797S mutation (Eberlein et al., 2015, Thress et al., 2015). Importantly, the authors demonstrate that the 3XmAbs strategy is capable of overcoming osimertinib resistance driven by the C797S mutation. This study has important medical implications for salvage therapy in patients who progress on osimertinib therapy. This manuscript is suitable for EMBO Molecular Medicine pending the authors addressing the following issues.

1. In Fig. EV1G and H, the authors provide evidence that the mice are able to tolerate the 3xmABs and osimertinib as monotherapy. Can the authors comment on the body weight, Fat Mass, and LEN mass of the animals treated with both osimertinib and 3XmAbs? This is in relation to practical concerns of toxicities associated with combination therapy of antibodies and small molecule inhibitors.
2. Fig. 2B - Provide the full time-course for the Caspase 3 cleavage western blot upon treatment with 3XmAb not just 48 hours.
3. Fig 3B and C - X-Gal and blot after Osimertinib treatment is missing. What are the levels of senescence markers in Osimertinib-treated cells compare to 3XmAbs treatment?
4. This manuscript would really benefit from a cartoon diagram summarising the different signalling, phenotypic and immunological effects of the 3XmAbs versus osimertinib.
5. The major limitation of this manuscript is that the majority of the studies are done on the PC9 model with some additional work in the H1975 system. While PC9 is one of the most well-studied models in the EGFR lung cancer field, the authors should caveat their findings within the context of this single model in the discussion. Can the authors also discuss the implications of their findings in relation to the recent studies demonstrating that 1G EGFRi resistant PC9 cells are composed of a heterogeneous population of pre-existing resistant clones and persister cells (Hata et al., 2016 and Ramirez et al., 2016)

Minor Issues

1. Fig 4 - the order of the panels are difficult to follow. Re-order panels to improve the flow.
2. Page 3 - introduction section. "While in animal models both drugs effectively inhibited growth of T790M-positive tumors....." This sentence is confusing as it suggests that 3XmAbs is a single drug. It needs further clarification.

Referee #2 (Comments on Novelty/Model System):

This AB combination is of interest for overcoming TKI resistance patients with advanced EGFR mutation-positive non-small cell lung cancer.

Referee #2 (Remarks):

EGFR tyrosine kinase inhibitors have been established as standard therapy for patients with advanced EGFR mutation-positive non-small cell lung cancer. Most recently, the third generation

EGFR tyrosine kinase inhibitor osimertinib has become standard treatment for patients who had developed T790M-mediated resistance during or after treatment with first- or second-generation EGFR tyrosine kinase inhibitors. While patients show good responses to these drugs, patients will eventually develop resistance against these drugs and will die from their disease. Therefore, strategies either to avoid the occurrence of resistance or to overcome resistance are of major clinical importance. In the present study, the authors demonstrated that a combination of three monoclonal antibodies can overcome resistance to EGFR tyrosine kinase inhibitors in pre-clinical studies. The antibody combination consisted of a monoclonal antibody directed against EGFR, an antibody against HER2 and one against HER3. This antibody combination was also studied in combination with osimertinib.

The authors demonstrated that the addition of an anti-HER3 antibody to cetuximab plus trastuzumab augmented their activity in cell lines and in xenograft models.

The authors demonstrated that their combination of three monoclonal antibodies had activity similar to the one of osimertinib. These activities were shown both *in vitro* and in mice models. The authors also demonstrated that the mechanism of resistance reversal by the antibody combination differed from the one of osimertinib. Osimertinib was shown to result in apoptosis. The antibody combination resulted in the down-regulation of the surface receptors for these antibodies and also induced senescence.

The antibody combination resulted in tumor shrinkage in tumor-bearing mice and this effect was stronger when the antibodies were combined with low doses of osimertinib. The antibody combination also prevented tumor recurrence and this effect was particularly seen when low doses of osimertinib had been added to the antibody combination. Finally, tumors which had become resistant to osimertinib through the acquisition of the C797S mutation were shown to remain sensitive to the antibody combination.

In conclusion, the findings of the present study are of importance and suggest a novel strategy to overcome resistance to EGFR tyrosine kinase inhibitors (including osimertinib) in patients with advanced EGFR mutation-positive non-small cell lung cancer. The paper is well written.

Minor comment

In the Discussion, the authors should add the results of the in the meantime published AURA3 phase 3 trial.

Referee #3 (Remarks):

The study by Mancini and collaborators evaluate the efficacy of a combination of antibodies anti-EGFR, Her2 and Her3 and osimertinib in cell lines harboring T790M mutations and C797S EGFR mutations. Authors use two cell line models to demonstrate that their combination of antibodies is active against different EGFR mutations. The main strength of the manuscript is that authors have performed a well conducted model of resistance to EGFR inhibitors and their pool of antibodies provide significant control of tumor growth in preclinical models. The weakness of the study is the demonstration of the mechanism of action; particularly in the case of immune system cells, the selected model is not appropriate.

Major comments:

1. The first part of the results "Combining trastuzumab and cetuximab..." is not clearly explained. Figures of both cell lines used are mixed, and no results of the *in vivo* model with the combination of the three antibodies are shown. It would be better to show results of each cell line in the same panel.
2. The demonstration of the protein expression of EGFR, Her2 and 3 in the membrane of the cells is crucial. It is only evaluated in some experiments, but it should be done in all of them. In the case of xenografts, in the case that FACS technique doesn't work in fresh tissue, The expression of these receptors could be detected by IHC.
3. As I have mentioned before, results of NK and macrophages have been performed in mice lacking T cells. In my opinion, this model is not suitable to obtain conclusions about interactions with immune system cells.
4. The effects on proliferation, apoptosis and senescence of all treatments are really interesting. However, these mechanisms have not been extensively evaluated in all models presented herein. Only ki67 is evaluated in figure 5, and senescence is evaluated in figure 7. The manuscript would be of greater interest if these mechanisms are evaluated in all preclinical models used.

Minor comments:

Reference citation: need to follow the journal specifications: "i.e. Smith & Jones, 2003; Smith et al, 2000"

1st Revision - authors' response

23 September 2017

Referee #1 (Remarks):

Mancini et al. examined the effect of a triple mixture of monoclonal antibodies targeting EGFR, HER2 and HER3 (3XmAbs), on first-generation EGFR inhibitor (1G EGFRi) resistant lung cancer models and compared their anti-cancer properties with the third-generation (3G) EGFR inhibitor osimertinib. The authors show that osimertinib and 3XmAbs comparably inhibit 1G EGFRi-resistant cell lines in vitro and in vivo. However, while osimertinib induces extensive apoptosis in 1G EGFRi-resistant tumours, 3XmAbs induce senescence. Interestingly, tumour relapse is prevented in tumour-bearing animals treated with a combination of 3XmAbs and osimertinib which is likely due to the distinct mechanisms of action of each drug. Previous studies have shown that patients treated with 3G EGFRi such as osimertinib acquire resistance, mostly due to the emergence of C797S mutation (Eberlein et al., 2015, Thress et al., 2015). Importantly, the authors demonstrate that the 3XmAbs strategy is capable of overcoming osimertinib resistance driven by the C797S mutation. This study has important medical implications for salvage therapy in patients who progress on osimertinib therapy. This manuscript is suitable for EMBO Molecular Medicine pending the authors addressing the following issues.

1. In Fig. EV1G and H, the authors provide evidence that the mice are able to tolerate the 3xmAbs and osimertinib as monotherapy. Can the authors comment on the body weight, Fat Mass, and LEN mass of the animals treated with both osimertinib and 3XmAbs? This is in relation to practical concerns of toxicities associated with combination therapy of antibodies and small molecule inhibitors.

As requested, we present in Figure I (below) the results of the analyses indicated by the Referee. As shown, we observed no evidence of additive or any potential toxicity when the three antibodies and the small molecule were applied together. It is notable, however, that the relatively young animals we used gained weight in the course of the experiment. A small delay in body weight gain, which did not reach statistical significance, was observed in the two cohorts treated with osimertinib, either alone or in combination with 3XmAbs. In addition, it is important noting that each of the three antibodies we tested recognizes the cognate human receptor with no known cross-reactivity toward the murine receptor. Note that Figure I has been added to the revised manuscript as Figs. EV4a and EV4b.

Figure I: Neither osimertinib nor 3XmAbs, or the combination of treatments, associate with marked effects on body weight or fat/lean parameters. (A) Comparison of body weights (averages \pm S.D.) of groups of eight CD1-nu/nu mice harboring PC9ER xenografts, which were treated with either erlotinib (50 mg/kg/dose), osimertinib (High; 5 mg/kg/dose), a mixture of three mAbs (3XmAbs; cetuximab, trastuzumab and mAb33; 0.2 mg/mouse/dose), or a combination of osimertinib (H) and 3XmAbs. Note that TKIs were daily administered using oral gavage, while the triple antibody combination was

injected intraperitoneally once every three days. (B) Shown are results of body mass composition analyses (mean \pm S.D.) of the fraction of fat mass (left) and lean mass (right) on day 20 of treatment. Mice harboring no tumors were used as an internal control.

2. Fig. 2B - Provide the full time-course for the Caspase 3 cleavage western blot upon treatment with 3XmAbs not just 48 hours.

As requested, we performed an experiment that examined potential cleavage of caspase 3 following prolonged incubation with 3XmAbs. The results obtained are shown in Figure II. Essentially, PC9ER cells were treated for up to 48 hours with 3XmAbs, but no cleaved forms of caspase 3 were observed at any time point. In contrast, treatment with osimertinib resulted in a clear signal. Notably, this result is consistent with several other lines of evidence, including annexin V, caspase 9 assays, as well as cleavage of BIM and PARP. Note that Fig. II has been added to Fig. 2B of the revised manuscript.

Figure II: Unlike osimertinib, which induces apoptosis of erlotinib-resistant NSCLC cells, no apoptosis is associated with 3XmAbs. PC9ER cells were treated for the indicated time intervals with 3XmAbs (CTX, TRZ and mAb33, each at 20 μ g/ml). Alternatively, cells were treated for 48 hours with osimertinib (0.5 μ M). Cell extracts were prepared, electrophoresed and immunoblotted for caspase 3 and its cleaved forms. GAPDH was used as an equal loading control. The locations of cleaved forms of caspase 3 are indicated. Blots are representative of two experiments.

3. Fig 3B and C - X-Gal and blot after Osimertinib treatment is missing. What are the levels of senescence markers in Osimertinib-treated cells compare to 3XmAbs treatment?

Original Figure 3B presented β -galactosidase (β -Gal) staining of PC9 cells, which were pre-treated for 11 days with 3XmAbs (or saline). Similarly, original Figure 3C presented immunoblots of whole extracts isolated from PC9ER cells that were pre-exposed for 2-8 days to 3XmAbs (CTX, TRZ and mAb33). Unfortunately, because already after 48 hours of treatment with osimertinib most cells (>75%) underwent apoptosis, or were already dead, we could not technically examine the senescence marker in extracts derived from osimertinib-treated cells. Nevertheless, to address this comment we stained the very few cells that survived osimertinib treatment (< 0.1%). As shown in Figure III, after staining for β -galactosidase (β -Gal) we noted that the majority of surviving cells were β -Gal positive. This observation confirmed that osimertinib/AZD9291 robustly induces apoptosis. However, a minor fraction of cells that resisted killing apparently acquired a state of senescence, in line with a secondary, senescence-inducing activity of the kinase inhibitor.

Figure III: Cells surviving apoptosis induced by a kinase inhibitor (osimertinib/AZD9291) exhibit positive beta-Gal staining. β -galactosidase (β -Gal) staining of PC9ER cells pre-treated for 14 days with 3XmAbs (20 μ g/ml), osimertinib (0.1 μ M), the respective combination (3XmAbs + osimertinib) or with saline (CTRL). Scale bar, 200 μ m.

4. This manuscript would really benefit from a cartoon diagram summarising the different signalling, phenotypic and immunological effects of the 3XmAbs versus osimertinib. *As requested, we added a cartoon diagram, which is presented below, to the revised manuscript.*

Figure IV: A cartoon diagram of anti-EGFR treatments relevant to NSCLC. The

model compares the three treatments we studied and refers to potential modes of action. EGFR (red), HER2 (green) and HER3 (blue) are shown as transmembrane molecules. Osimertinib is represented by the letter X, which blocks the intracellular kinase domain of EGFR. Antibodies to EGFR, HER2 and HER3 are schematically shown. Note that the combination of the three antibodies (3XmAbs) and osimertinib (AZD9291) makes use of a relatively low dose of the kinase inhibitor (broken line X). This combination of drugs, as well as 3XmAbs alone, overcomes both resistance-causing mutations, namely T790M and C797S, and it persistently prevents tumor relapses.

5. The major limitation of this manuscript is that the majority of the studies are done on the PC9 model with some additional work in the H1975 system. While PC9 is one of the most well-studied models in the EGFR lung cancer field, the authors should caveat their findings within the context of this single model in the discussion. Can the authors also discuss the implications of their findings in relation to the recent studies demonstrating that 1G EGFRi resistant PC9 cells are composed of a heterogeneous population of pre-existing resistant clones and persister cells (Hata et al., 2016 and Ramirez et al., 2016).

As requested, we refer in the revised manuscript, especially in the re-written Discussion, to the caveat of using only two model cancer lines. In addition, we now refer to the possibility raised by recent studies, which implicated pre-existing mutant clones, as well as newly emerging mutations, in delayed resistance to second-, and maybe also to third-generation kinase inhibitors. Within this context, we refer to the two 2016 reports indicated by the Referee.

Minor Issues

1. Fig 4 - the order of the panels are difficult to follow. Re-order panels to improve the flow.

As requested, we rearranged many panels of the revised manuscript in a way that improves the flow and increases clarity.

2. Page 3 - introduction section. "While in animal models both drugs effectively inhibited growth of T790M-positive tumors....." This sentence is confusing as it suggests that 3XmAbs is a single drug. It needs further clarification.

As requested, we re-wrote the sentence such that the revised text avoids confusion.

Referee #2 (Comments on Novelty/Model System):

This AB combination is of interest for overcoming TKI resistance patients with advanced EGFR mutation-positive non-small cell lung cancer.

Referee #2 (Remarks):

EGFR tyrosine kinase inhibitors have been established as standard therapy for patients with advanced EGFR mutation-positive non-small cell lung cancer. Most recently, the third generation EGFR tyrosine kinase inhibitor osimertinib has become standard treatment for patients who had developed T790M-mediated resistance during or after treatment with first- or second-generation EGFR tyrosine kinase inhibitors. While patients show good responses to these drugs, patients will eventually develop resistance against these drugs and will die from their disease. Therefore, strategies either to avoid the occurrence of resistance or to overcome resistance are of major clinical importance. In the present study, the authors demonstrated that a combination of three monoclonal antibodies can overcome resistance to EGFR tyrosine kinase inhibitors in pre-clinical studies. The antibody combination consisted of a monoclonal antibody directed against EGFR, an antibody against HER2 and one against HER3. This antibody combination was also studied in combination with osimertinib.

The authors demonstrated that the addition of an anti-HER3 antibody to cetuximab plus trastuzumab augmented their activity in cell lines and in xenograft models.

The authors demonstrated that their combination of three monoclonal antibodies had activity similar to the one of osimertinib. These activities were shown both in vitro and in mice models. The authors also demonstrated that the mechanism of resistance reversal by the antibody combination differed from the one of osimertinib. Osimertinib was shown to result in apoptosis. The antibody combination resulted in the down-regulation of the surface receptors for these antibodies and also

induced senescence.

The antibody combination resulted in tumor shrinkage in tumor-bearing mice and this effect was stronger when the antibodies were combined with low doses of osimertinib. The antibody combination also prevented tumor recurrence and this effect was particularly seen when low doses of osimertinib had been added to the antibody combination. Finally, tumors which had become resistant to osimertinib through the acquisition of the C797S mutation were shown to remain sensitive to the antibody combination.

In conclusion, the findings of the present study are of importance and suggest a novel strategy to overcome resistance to EGFR tyrosine kinase inhibitors (including osimertinib) in patients with advanced EGFR mutation-positive non-small cell lung cancer. The paper is well written.

Minor comment

In the Discussion, the authors should add the results of the in the meantime published AURA3 phase 3 trial.

As requested, the revised Discussion refers to the AURORA3 trial.

Referee #3 (Remarks):

The study by Mancini and collaborators evaluate the efficacy of a combination of antibodies anti-EGFR, Her2 and Her3 and osimertinib in cell lines harboring T790M mutations and C797S EGFR mutations. Authors use two cell line models to demonstrate that their combination of antibodies is active against different EGFR mutations. The main strength of the manuscript is that authors have performed a well conducted model of resistance to EGFR inhibitors and their pool of antibodies provide significant control of tumor growth in preclinical models. The weakness of the study is the demonstration of the mechanism of action; particularly in the case of immune system cells, the selected model is not appropriate.

Major comments:

1. The first part of the results "Combining trastuzumab and cetuximab..." is not clearly explained. Figures of both cell lines used are mixed, and no results of the in vivo model with the combination of the three antibodies are shown. It would be better to show results of each cell line in the same panel.

In response to this comment, we performed a major revision of the first part of the manuscript. In other words, we combined the results related to treatment with 3XmAbs in a new Figure (see new Figure 1A), and revised the respective text. In parallel, the corresponding supplementary figure (Figure EV1) assembles now all in vitro assays relevant to the revised form of Figure 1A. In addition, to increase clarity, we indicated within each panel the identity of the cell line we used (i.e., PC9ER or H1975). Likewise, wherever possible we show in the revised manuscript the results obtained from the two cell models, one next to the other.

2. The demonstration of the protein expression of EGFR, Her2 and 3 in the membrane of the cells is crucial. It is only evaluated in some experiments, but it should be done in all of them. In the case of xenografts, in the case that FACS technique doesn't work in fresh tissue. The expression of these receptors could be detected by IHC.

As requested, the revised manuscript presents several analyses of receptor expression. The list of figures and the corresponding methods is shown below.

- Figure 1C: Western blot analyses of the three receptors following treatments with increasing concentrations of osimertinib, erlotinib, CO-1686 and 3XmAbs. As shown, the abundance of each of the three receptors (EGFR, HER2 and HER3) underwent downregulation in cells treated with the triple antibody combinations, but neither third-generation kinase inhibitor similarly acted. In fact, these inhibitors either stabilized or enhanced receptor expression.

- Figure 5A: Western blot analysis of the three receptors following treatment with osimertinib (10 and 100 nM) or a combination of osimertinib plus 3XmAbs. As shown, treatment with the kinase inhibitor enhanced expression levels, but the combination of antibodies downregulated levels of expression, especially the abundance of HER2 and HER3.

- Figure 5B: FACS analyses of surface levels of the three receptors, showing that 3XmAbs almost completely removed all three receptors from the cell surface, whereas osimertinib alone increased surface expression (especially surface levels of HER3), whereas the combination of osimertinib and 3XmAbs exerted a similar effect to that induced by 3XmAbs only.

- Figure EV1c: FACS analyses of surface expression of the three receptors following treatment with

single antibodies, and two different third-generation kinase inhibitors. In addition to demonstrating downregulation of all three receptors by the triple antibodies (two different concentrations), the results confirmed that two different antibodies to EGFR, when applied alone, almost completely cleared EGFR from the surface of lung cancer cells.

- Figure EV1e: IHC analyses of H1975 xenografts for EGFR and phosphor-EGFR, confirming the ability of 3XmAbs to downregulate EGFR, along with pEGFR. By contrast, while osimertinib reduced pEGFR signals, erlotinib was ineffective and neither mimicked the effect of 3XmAbs on total EGFR levels.

To directly address the Referee's comment and also expand the spectrum of receptor expression assays, we combined IHC and immunofluorescence on thin slices (6 μm) of H1975 NSCLC xenografts, which were treated for 10 days prior to excision and fixation in formalin. The results are presented below (Figure V) and they reflect downregulation of EGFR and HER2, as well as weaker downregulation of HER3, after treatment with 3XmAbs (or with the combination with osimertinib), but neither erlotinib nor osimertinib exerted a similar effect. Note that the new data has been inserted in the revised manuscript (Figure 5C).

Figure V: In vivo downregulation of EGFR, HER2 and HER3 by a combination of three antibodies. CD1 nu/nu mice harboring H1975 xenografts were treated daily with erlotinib (50 mg/kg/day) and osimertinib (5 mg/kg/day) using oral gavage, while the triple antibody combination (3XmAbs; CTX, TRZ and mAb33; 0.2 mg/mouse/injection) and saline (vehicle) were administered intraperitoneally once every three days, as indicated (for 10 days). Thereafter, tumors were harvested, formalin fixed and paraffin-embedded. The corresponding sections, obtained from each tissue block, were analyzed by immunofluorescence using antibodies specific to EGFR (Cell signaling, #4267S), HER-2 (Cell Signaling, #4290S) and HER-3 (Cell Signaling #12708S). Stained sections were examined and photographed using a fluorescence microscope (Eclipse Ni-U, Nikon, Tokyo, Japan) equipped with a Plan Fluor objectives (20X) connected to a monochrome camera (DS-Qi1, Nikon). Scale bar 25 μm . Also

shown are histograms depicting quantifications of receptor staining using 3-6 section/tumor. *, $p < 0.05$; ***, $p < 0.001$; ****, $p < 0.0001$; one-way ANOVA with Tukey's test.

3. As I have mentioned before, results of NK and macrophages have been performed in mice lacking T cells. In my opinion, this model is not suitable to obtain conclusions about interactions with immune system cells.

This comment is in line with the Editor's summary comment that reads as follows:

"...but there was also a clear consensus that we would not be asking you to address the immune aspect. It was actually suggested that it would be altogether better to remove that part from the manuscript to improve both its focus and clarity."

In response to these two comments, we removed from the revised manuscript all data related to potential immune mechanisms acting in vivo, as well as relevance to specific populations of myeloid or lymphoid cells. These aspects of the treatments we compared will be the subjects of our future study. Notably, however, data related to the ability of the antibodies to induce ADCC in vitro, as well as the ability of both treatments to recruit unidentified hematopoietic cells to xenografts (Figs. EV3g) were retained in the revised manuscript.

4. The effects on proliferation, apoptosis and senescence of all treatments are really interesting. However, these mechanisms have not been extensively evaluated in all models presented herein. Only ki67 is evaluated in figure 5, and senescence is evaluated in figure 7. The manuscript would be of greater interest if these mechanisms are evaluated in all preclinical models used.

As requested, we extended the analysis of cellular responses to both tumor models employed by our study. The results are shown and listed below.

(i) *We used KI67 as a marker of cell proliferation taking place within xenografts (in vivo). The results summarized below indicate that the two xenograft models we used similarly responded to osimertinib and 3XmAbs.*

- *Figure VI (below) presents KI67 staining of H1975 tumors grown in mice. As shown, both osimertinib and 3XmAbs comparably inhibited cell proliferation, whereas erlotinib was ineffective, probably due to the T790M mutation of EGFR. Note that the data shown in Figure VI has been added to the revised manuscript (Fig. 1E and EV1d).*
- *Figures 4C and 4D of the revised manuscript present IHC staining of PC9ER xenografts, showing that the combination of 3XmAbs and low dose osimertinib significantly inhibited tumor cell proliferation compared to high dose treatment with osimertinib.*
- *Figure VII (below) presents results of KI67 staining of slices obtained from xenografts of lung cancer cells that acquired resistance to osimertinib (PC9ER-AZDR cells). Evidently, while osimertinib was ineffective, treatment of mice with 3XmAbs revealed that osimertinib-resistant tumors could still be inhibited by the triple mixture of antibodies (3XmAbs). Note that the data shown in Figure VII has been added to the revised manuscript (Figs. 6G and 6H).*

Figure VI: Both osimertinib and the triple antibody mixture inhibit proliferation of erlotinib resistant cells. H1975 cells (3×10^6 cells per animal) were subcutaneously grafted in the flanks of CD1-nu/nu mice, which were subsequently randomized and subjected to the

following treatments: erlotinib (50 mg/kg/day), osimertinib (5 mg/kg/day) or 3XmAbs (CTX, TRZ and mAb33; 0.2 mg/mouse/injection; administered twice a week). Shown is immunohistochemical staining for KI67 in paraffin-embedded sections using specific antibodies. Scale bars, 100 μ m. Also shown quantifications of KI67 staining using 6-8 sections/tumor. ***, $p < 0.001$; ****, $p < 0.0001$; one-way ANOVA with Tukey's test.

Figure VII: Unlike osimertinib, the triple antibody mixture inhibits proliferation of xenografts derived from osimertinib-resistant lung cancer cells (PC9ER-AZDR). PC9ER-AZDR cells (3×10^6 cells per animal) were subcutaneously grafted in CD1-nu/nu mice. Thereafter, tumor-bearing animals were randomized into groups of 9-10 mice, which were later treated once every 3 days with 3XmAbs (0.2 mg/mouse/injection). Alternatively, mice were orally treated with two doses of osimertinib (1 or 5 mg/kg/dose; L or H, respectively). Shown is immunohistochemical staining for KI67 in paraffin-embedded sections. Scale bars, 100 μ m. Also shown quantifications of KI67 staining using 6-8 sections/tumor. ***, $p < 0.001$; ****, $p < 0.0001$; one-way ANOVA with Tukey's test.

(ii) As requested, the revised manuscript presents side-by-side comparisons showing the results of cell death assays, namely caspase 3 cleavage, performed with the two xenograft models (PC9ER and H1975; see Figure 2E of the revised manuscript and Figure VIII, below). Briefly, two weeks after inoculation of the respective cell lines, mice were randomized (3-4 mice/group) and treated for 12 days either with vehicle, 3XmAbs (CTX, TRZ and mAb33; 0.2 mg/mouse/injection, once every three days) or with osimertinib (5 mg/kg/day, oral administration). Immunohistochemical staining for cleaved caspase 3 was performed on paraffin-embedded sections derived from the two types of xenografts. Quantification of the results obtained in three different experiments confirmed the ability of osimertinib to induce strong apoptosis in both tumor models ($p < 0.001$; two-way analysis of variance, ANOVA, with Tukey's comparison). Consistent with other assays, the apoptosis signal observed when applying 3XmAbs were very weak, and in the H1975 model they did not reach statistical significance.

Figure VIII: Unlike osimertinib, which increases apoptosis in tumor xenografts, 3XmAbs is a weak inducer of cell death. Immunohistochemical staining for cleaved caspase 3 performed on paraffin-embedded sections derived from xenografts of either PC9ER or H1975 cells. Two weeks after tumor inoculation, mice were randomized (3-4 mice/group) and treated for 12 days either with vehicle, 3XmAbs (0.2 mg/mouse/injection, once every three days) or osimertinib (5 mg/kg/daily). Scale bars, 100 μm. Also shown is a quantification of the results obtained in three different experiments. ****, $p < 0.0001$; ***, $p < 0.001$; *, $p < 0.01$ (two-way analysis of variance, ANOVA, with Tukey's comparison).

Minor comments:

Reference citation: need to follow the journal specifications: "i.e. Smith & Jones, 2003; Smith et al, 2000"

As requested, all reference citations were revised per the "Instructions for Authors".

2nd Editorial Decision

10 October 2017

Thank you for the submission of your revised manuscript to EMBO Molecular Medicine. We have now received the enclosed reports from the referees that were asked to re-assess it. As you will see the reviewers are now supportive and I am pleased to inform you that we will be able to accept your manuscript pending editorial final amendments.

***** Reviewer's comments *****

Referee #1 (Remarks for Author):

The authors have addressed all my comments in this revision.

Referee #3 (Comments on Novelty/Model System for Author):

In the revised submission, Mancini et al have adequately addressed all the concerns raised in my initial review.

Corresponding Author Name: Yosef Yarden

Journal Submitted to: EMM

Manuscript Number: EMM-2017-08076